


# Retrieval of microphysical dust particle properties from SALTRACE lidar observations: Case studies

Stefanos Samaras[1], Christine Böckmann[2], Moritz Haarig[3], Albert Ansmann[3], Adrian Walser[4], and Bernadett Weinzierl[4]

[1]German Aerospace Center, German Remote Sensing Data Center Atmosphere, Oberpfaffenhofen, Germany
[2]Institute of Mathematics, University of Potsdam, Potsdam - OT Golm, Germany.
[3]Leibniz Institute for Tropospheric Research, Leipzig, Germany.
[4]Department of Aerosol Physics and Environmental Physics, University of Vienna, Vienna, Austria

*Correspondence to:* Christine Böckmann (Christine.Boeckmann@uni-potsdam.de)

**Abstract.** Saharan dust is a major natural atmospheric aerosol component with significant impact on the Earth radiation budget. In this work we determine the microphysical properties of dust particles after a long-range transport over the Atlantic Ocean, using input from three depolarization channels of a multi-wavelength polarization Raman lidar. The measurements were performed at Barbados in the framework of the Saharan Aerosol Long-Range Transport and Aerosol-Cloud-Interaction Experiment (SALTRACE) in the summers of 2013 and 2014. The microphysical retrievals are performed with the software tool SphInX (Spheroidal Inversion Experiments) which uses regularization for the inversion process and a new two-dimensional (2-D) extension of the Mie model approximating dust with spheroids. The method allows us to simultaneously retrieve shape- and size-dependent particle distributions. Because dust particles are mostly non-spherical this software tool fills the gap in estimating the non-spherical particle fraction. Two cases measured on 10 July 2013 and 20 June 2014 are discussed. 2-D radius-bimodal shape-size distribution are retrieved. The ratio of spherical-to-non-spherical contributions to the particle number concentration was found to be about 3/7. A volume-weighted effective aspect ratio of 1.1 was obtained, indicating slightly prolate particles. The total effective radius for the two cases in the preselected radius range from 0.01-2.2 μm was found to be, on average, 0.75 μm. The stronger dust event (10 July 2013) showed about 24% higher values for the total surface-area and volume concentration. Finally, we compare our results with the ones from the polarization lidar-photometer networking (POLIPHON) method and ground-based photometers as well as with airborne in situ particle counters. Considering all differences in these independent approaches, we find a qualitatively good agreement between the different results and a consistent description of the dust cases. Such an extensive comparison is a novel and fruitful exercise and corroborates that the mathematical retrieval based on Raman lidar data of particle backscattering, extinction, and depolarization is a powerful tool even in the case of dust particles.

## 1 Introduction

In recent years, mineral dust has received growing interest in atmospheric and climate research because of its significant impact on cloud formation and the radiation budget of the Earth. It contributes to about 50% of the global annual particle emission by





mass, attributed mainly to North Africa (Huneeus et al. , 2011). Although important progress concerning the role of dust in the climate system has been made in the last 10 years many questions remain open (IPCC , 2019).

"New field campaigns, as well as new analyses of observations from prior campaigns, have produced insights into the role of dust in western Africa in climate system, such as long-ranged transport of dust across the Atlantic (Groß et al. , 2015) and the
characterisation of aerosol particles and their ability to act as ice and cloud condensation nuclei (Price et al. , 2018). Size distribution at emission is another key parameter controlling dust interactions with radiation. Most models now use the parametrisation of Kok  (2011) based on the theory of brittle material. It was shown that most models underestimate the size of the global dust cycle (Kok , 2011). Characterisation of spatial and temporal distribution of dust emissions is essential for weather prediction and climate projections (high confidence). Although there is a growing confidence in characterising the seasonality
and peak of dust emissions (i.e., spring–summer (Wang et al. , 2015)) and how the meteorological and soil conditions control dust sources, an understanding of long-term future dust dynamics, inter-annual dust variability and how they will affect future climate still requires substantial work. Dust is also important at high latitude, where it has an impact on snow-covered surface albedo and weather (Bullard , 2016).", from IPCC web page https://www.ipcc.ch/site/assets/uploads/sites/4/2019/11/05_Chapter-2.pdf, pages 166-167.

Particle size distribution and single scattering albedo are critical and important quantities in radiative forcing computations. In particular, the dust coarse mode fraction has strong impact on the radiative budget (Weinzierl et al. , 2017; Ryder et al. , 2019). The three field campaigns conducted in the framework of the Saharan Aerosol Long-Range Transport and Aerosol-Cloud-Interaction Experiment (SALTRACE) in the summer of 2013 and in the winter and summer of 2014 aimed to investigate the above-mentioned topics (Weinzierl et al. , 2017) and (http://www.pa.op.dlr.de/saltrace/index.html).

In this article, we apply a new inversion method applied to multiwavelength polarization Raman lidar observations of particle optical properties to derive the dust particle size distribution and microphysical properties of mineral dust after long-range transport from the Saharan desert across the Atlantic Ocean to the Caribbean island of Barbados. Part of the retrieval is the separation of fine and coarse mode dust. To solve the so-called ill-posed problem we use the inversion technique with regularization. Such methods are a widely used and permit an efficient retrieval of the size distribution for spherical particles
(Müller et al., 1999; Böckmann , 2001; Veselovskii et al. , 2004; Müller and Böckmann et al. , 2016). In Böckmann and Wauer  (2001), algorithms for spherically multilayered and absorbing particles are presented. Simulations show that the influence of particle shape properties and chemical composition is not negligible and need to be carefully considered when inversion algorithms are used to determine the particle microphysical properties.

Lopatin et al.  (2013) present the GARRLiC (Generalized Aerosol Retrieval from Radiometer and Lidar Combined data)
algorithm which assumes height-independent microphysical particle properties and permits retrievals of the vertical profiles of both fine and coarse mode aerosol concentrations as well as the size distribution and the complex refractive index for each mode.

In most attempts, the Mie scattering model is used. This model offers the computation of the optical properties of fine mode aerosol (Böckmann et al. , 2005) such as biomass burning smoke and urban haze with good accuracy (Osterloh et al. , 2013;
Samaras et al. , 2015a; Ortiz-Amezcua et al. , 2017; Ritter et al. , 2018). However, the Mie model no longer provides a viable



description of optical properties in the case of mineral dust particles since dust particles are clearly non-spherical in shape. The consideration of non-sphericity, even in its simplest form by assuming a spheroidal shape for the dust particles, adds multiple layers of complexity to the retrieval procedure. Numerical stability becomes an important issue as well.

Nonetheless, particle shape is known to have substantial impact on the scattering characteristics, especially with respect to backward scattering (Mishchenko et al. , 1996). The assumption of spheroidal particle geometry seems to reproduce the optical properties of dust particles significantly better than the use of spherical shape model (Mishchenko et al. , 1996; Kahnert and Kylling , 2004). Ensemble averages based on the T-matrix method for spheroids agree better with measurements than computations based on Mie theory (Bohren and Singham , 1991). Simulations with a mixture of simple spheroids are found to agree well with measured optical properties of non-spherical aerosol particles. The exact knowledge of the shape characteristics in the modeling efforts is obviously not needed in many cases of measurements with contributions from individual particles with different orientations, shapes, and composition (Wiscombe and Mugnai , 1986; Bohren and Singham , 1991; Mishchenko et al. , 1997; Kahnert and Kylling , 2004; Veselovskii et al. , 2010). Several non-spherical-particle approximations are present in the literature which focuses on optical parameter investigations. In contrast, non-sphericity is usually considered in the microphysic retrieval branch by using the spheroid shape model (Dubovik et al. , 2002; Veselovskii et al. , 2010; Osterloh , 2011; Böckmann and Osterloh , 2014; Samaras et al. , 2015b; Samaras , 2017).

The microphysical parameter database hosted by the Aerosol Robotic Network (AERONET) is such an example. Aerosol particle size distributions are assumed to consist of an ensemble of polydisperse, homogeneous spheres and a mixture of polydisperse, randomly-oriented homogeneous spheroids with size-independent aspect-ratio distribution (Giles et al. , 2019). A customary assumption in non-spherical microphysical retrieval codes, adopted also by AERONET, is that shape and size are independent parameters with the consequence that particle shape is given by a separate aspect ratio distribution. This approach was used by Dubovik et al. (2006) to invert AERONET sun-photometer data into microphysical particle properties.

Dubovik et al. (2006) show that mixtures of spheroids can well explain measured spectral and angular dependencies of scattering intensity and polarization. Furthermore, the fraction on non-sphercial particles for aerosol mixtures with a dominant coarse mode (particle radii $\geq \sim 1\mu$m) can be determined. The retrieval results indicate that nonspherical particles with aspect ratios $\sim 1.5$ and higher dominate in desert dust plumes. In Müller et al. (2013) this concept was then also used in lidar inversion procedures.

Bi et al. (2018) present the invariant imbedding T-matrix method to assess the backscattering linear depolarization ratio (LDR) of non-spherical particles in a super-ellipsoidal shape space. Super-ellipsoids have inherent flexibility regarding modelling of particle aspect ratio, roundness, and concavity, which are the salient characteristics of, e.g., sea salt and dust aerosols. The results provide comprehensive insight in the relationship between LDR of atmospheric aerosols and particle shape and refractive index characteristics. This can be used in the interpretation of lidar observations. Gasteiger et al. (2011) modelled lidar-relevant optical properties of mineral dust aerosol and compared the simulated results with optical properties derived from lidar data during the Saharan Mineral Dust Experiment (SAMUM) and found that irregularly shaped dust particles with typical refractive index characteristics cause a higher linear depolarization ratio than corresponding spheroids. Simulations with a variety of irregular-shape models improved the agreement with lidar observations.



In this work we are using the new concept of a two-dimensional (2-D) spheroidal particle model implicit in the software tool SphInX (Spheroidal Inversion eXperiments) developed by Samaras (2017). In contrast to Dubovik et al. (2006) and Müller et al. (2013), no restriction regarding shape-size separability is given. It is left to the retrieval to reveal the shape-size relationship. In fact, it appears that there is a link between the aspect ratio distributions and particle size as it was found

in independent studies during the SAMUM campaign (Kandler et al. , 2009; Koepke et al. , 2015). The SphInX tool was already successfully used in Soupiona et al. (2019) to investigate aerosol optical and microphysical properties during selected Saharan dust events over Athens (Greece; NE Mediterranean) and Granada (Spain; NW Mediterranean) focusing on short and long-range dust processes.

SphInX is able to include depolarization ratio information at any wavelength. Depolarization-ratio measurements with ad-

vanced polarization lidars at four wavelengths (355, 532, 710 and 1064 nm) (Freudenthaler et al. , 2009) were performed during SAMUM. The mean linear particle depolarization ratio at 532 nm was 0.31, in the range between 0.27 and 0.35 with a slight decrease of the linear particle depolarization ratio with wavelength between 532 and 1064 nm from 0.31 ± 0.03 to 0.27 ± 0.04. Other statistical properties of Saharan dust are provided in Tesche et al. (2009) during SAMUM. The 500 nm dust optical depth ranged from 0.2–0.8 at the field site south of the High Atlas mountains, Ångström exponents derived from photometer

and lidar data were between 0–0.4. On average, extinction-to-backscatter ratios of 53–55 sr (±7–13 sr) were obtained at 355, 532 and 1064 nm. The question arises which depolarization channel is the most valuable for the microphysical retrieval. It is not surprising that the answer depends on the aerosol type. Gasteiger and Freudenthaler (2014) found for transported Eyjaf-jallajökull volcano ash, e.g., that additional 1064 nm depolarization measurements significantly reduce the uncertainty in the retrieved mass concentration and effective particle size. This significant improvement in accuracy is the result of the increased

sensitivity of the measured optical data to the impact of larger particles.

Here we investigate two measurement cases from SALTRACE Barbados campaign. The measurements were conducted on 10 July 2013 (case 2) between 23:15-0:45 UTC and 20 June 2014 (case 1) between 23:10-02:10 UTC. The used lidar of the Leibniz Institute for Tropospheric Research (TROPOS), shortly called BERTHA (Backscatter Extinction lidar Ratio Temper-ature Humidity profiling Apparatus), provides depolarization ratio profiles at 355, 532 and 1064 nm (Haarig et al. , 2017a) in

addition to three backscatter and two extinction coefficient profiles, the so-called $3\beta + 2\alpha$ data set. These measurements were basically performed to characterize optical and microphysical properties of aged dust plumes after long-range transport across the Atlantic Ocean. Both cases were already investigated by Mamouri and Ansmann (2017); Haarig et al. (2017a).

Furthermore, case 1 was also investigated in Tesche et al. (2019) with focus on the retrieval of microphysical properties and the guiding question to what extent uncertainties can be reduced or avoided by including specific depolarization-ratio

information in the data analysis. In contrast to our approach, Tesche et al. (2019) used the model of Dubovik et al. (2006) with a fixed axis-ratio distribution. Tesche et al. (2019) found that the consideration of light-depolarization information strongly improves the quality of the retrieval products.

Case 2 provides a unique opportunity to compare our lidar inversion results (profiles) with airborne in situ particle counter measurements. In addition, we compare our results with respective ones obtained by means of the polarization lidar-photometer





networking (POLIPHON) method (Mamouri and Ansmann , 2017). Additional comparisons are made with inversion products from AERONET sun-photometer observations.

The structure of this paper will be as follows. In Section 2, we provide an overview of the 2-D model methodology and the used instruments. Section 3 presents the results of the two case studies. The final Section 4 summarizes our findings and
contains concluding remarks.

## 2 Preliminaries

### 2.1 2-D Model and Algorithm

The extinction- ($\alpha$) and backscatter coefficients ($\beta$), both denoted with $\mathcal{Z}(\lambda)$ with wavelength $\lambda$ are commonly defined as the collective interaction (scattering/absorption) probability over a particle number size distribution (PSD) $n(r)$. Denoting with
$C_{\alpha/\beta}(r, \lambda; m)$ the associated (interaction) cross sections, we can express this relation through a Fredholm integral of the first kind

$$\mathcal{Z}(\lambda) = \int\limits_{r_{\min}}^{r_{\max}} C_{\alpha/\beta}(r, \lambda; m)\, n(r)\mathrm{d}r, \tag{1}$$

where $m$ is the complex refractive index (RI). A Fredholm integral equation of first kind is a compact operator and therefore the inversion of it is an ill-posed problem. We refer to Eq. 1 shortly as the Mie model, since the optical efficiencies are calculated
by the (Lorenz-) Mie theory.

We extend this model considering a spheroid-particle approximation, which needs only one additional parameter, i.e. the aspect ratio $\eta$, to be fully defined. It is formalized as follows:

$$\mathcal{Z}(\lambda) = \int\limits_{\eta_{\min}}^{\eta_{\max}} \int\limits_{r_{\min}}^{r_{\max}} C_{\alpha/\beta}(r, \lambda, \eta; m)\, n(r, \eta)\mathrm{d}r\mathrm{d}\eta, \tag{2}$$

where $[r_{\min}, r_{\max}]$ and $[\eta_{\min}, \eta_{\max}]$ are sensible and sufficiently wide radius- and aspect ratio ranges determined experimentally
as part of the initial parameters of a microphysical retrieval as discussed in Samaras et al. (2015b). Thus, the integral over size, i.e. the radius $r$, is integrated again over a parameter representing the different aspect ratios $\eta$ of spheroids, i.e. oblate- ($\eta < 1$), spherical- $\eta = 1$, and prolate ($\eta > 1$) shapes. This means that we are now looking for 2-D particle number shape-size distributions. Note that there is no particular shape-size separation relation for the shape-size distribution, as for instance in Veselovskii et al. (2010). The radius here makes sense as the one of a sphere with equal volume to a spheroid. The latter is
found by $V = 4\pi a^2 b/3$, so that $r = \sqrt[3]{a^2 b}$, where $a$ and $b$ are the semi-minor and semi-major axis of the spheroid, respectively. Regarding the PSD, we see that the case of spheroids defines a two-dimensional (2-D) generalization of the spherical model, but the optical parameters $\mathcal{Z}$ still remain only wavelength-dependent, which is why we shall consider this, from a mathematical point of view, a quasi-2-D model. Particle orientation comes about naturally when addressing non-sphericity but is generally unknown. For this, we further assume randomly oriented spheroidal particles for the calculations of the scattering cross sections





in Eq. 2 providing further simplifications to the model. Replacing the scattering cross sections with dimensionless efficiencies $Q_{\alpha/\beta}$ and the number- with volume distribution we can reformulate the generalized model Eq. 2 to the one we use in practice for our application

$$\mathcal{Z}(\lambda) = \int\limits_{\eta_{\min}}^{\eta_{\max}} \int\limits_{r_{\min}}^{r_{\max}} \frac{3A}{16\pi r^3} Q_{\alpha/\beta}(r,\lambda,\eta;m)\, v(r,\eta)\mathrm{d}r\mathrm{d}\eta, \tag{3}$$

where $A$ is the particle surface area, where we used the fact that in a convex particle ensemble the average area per particle is equal to $A/4$ (van de Hulst , 1958).

A precalculated database will be used, created by the software tool Mieschka (Rother and Kahnert , 2014). Mieschka is able to perform scattering-related T-matrix calculations for spherical particles and rotationally symmetric non-spherical particles with a convergence strategy based on Barber and Hill  (1990); Wiscombe and Mugnai  (1986). Additionally, it provides an
extensive database of scattering quantities for spheroidal geometries. Mieschka's look-up tables include scattering efficiencies for a $6 \times 7$, $(\Re(m) \times \Im(m))$ refractive index grid, a total of 42 RI values, 7 different aspect ratios and a size parameter range [0.02, 40 $\mu$m] with a resolution of 0.2 $\mu$m, see Table 1. While the maximum size parameter is reasonably large, its potential cannot be fully exploited here in terms of the radius extent since the lidar wavelength at 355 nm restricts the maximum radius ($r_{\max}$) to about 2.2$\mu$m. All working formulas are with respect to $r$. The resolution gap in the aspect ratio needed for the
integration is handled by interpolation to the nearest neighbor.

The microphysical parameters can be redefined in 2-D analogously. Similarly to the 1-D case we set the number shape-size distribution $n(r,\eta) = \frac{3}{4\pi r^3}v(r,\eta)$, where $v(r,\eta)$ is the volume shape-size distribution. We define the number concentration

$$n_{\mathrm{t}} = \int\limits_{\eta_{\min}}^{\eta_{\max}} \int\limits_{r_{\min}}^{r_{\max}} n(r,\eta)drd\eta, \tag{4}$$

the total volume concentration

$$v_{\mathrm{t}} = \int\limits_{\eta_{\min}}^{\eta_{\max}} \int\limits_{r_{\min}}^{r_{\max}} v(r,\eta)drd\eta, \tag{5}$$

and the surface-area concentration

$$a_{\mathrm{t}} = \int\limits_{\eta_{\min}}^{\eta_{\max}} \int\limits_{r_{\min}}^{r_{\max}} \frac{3}{\pi r^3}G(r,\eta)v(r,\eta)drd\eta. \tag{6}$$

The effective radius is calculated by $r_{\mathrm{eff}} = 3\frac{v_t}{a_t}$ , using Eq. 5 and 6. The function $G(r,\eta)$ denotes here the spheroidal geometrical cross section of the particle, which can be explicitly computed as follows

$$G(r,\eta) = \begin{cases} 2\pi\left[a^2 + \frac{b^2}{e}\tanh^{-1}(e)\right], & \text{where } e = \sqrt{1-b^2/a^2}, & \text{if } \eta < 1, \\ 4\pi r^2, & & \text{if } \eta = 1, \\ 2\pi\left[a^2 + \frac{ab}{e}\sinh^{-1}(e)\right], & \text{where } e = \sqrt{1-a^2/b^2} & \text{if } \eta > 1. \end{cases} \tag{7}$$





Furthermore some new parameters are introduced (Samaras , 2017) in order to study the shape of the size distribution in more detail. We define the volume-weighted- effective aspect ratio

$$\eta_{\mathrm{eff}} = \frac{\int_{\eta_{\min}}^{\eta_{\max}} \eta \int_{r_{\min}}^{r_{\max}} v(r,\eta) dr d\eta}{v_{\mathrm{t}}}, \tag{8}$$

and the aspect ratio width

$$\eta_{\mathrm{var}} = \frac{\int_{\eta_{\min}}^{\eta_{\max}} (\eta - \eta_{\mathrm{eff}})^2 \int_{r_{\min}}^{r_{\max}} v(r,\eta) dr d\eta}{v_{\mathrm{t}}}. \tag{9}$$

The parameters $\eta_{\mathrm{eff}}$ and $\eta_{\mathrm{var}}$ give us an estimation of a central tendency of the aspect ratio and the spread of the values from this type of the mean.

The following parameter specializes in identifying individual (non-)spherical contributions (%) in the volume concentration. We define the non-spherical volume concentration fraction by

$$\gamma = 1 - \frac{1}{v_{\mathrm{t}}} \int_{1-\chi}^{1+\chi} \int_{r_{\min}}^{r_{\max}} v(r,\eta) dr d\eta, \tag{10}$$

where $\chi$ is a positive small, yet not negligible number called the shape-transition parameter. Similarly we call $1-\gamma$ the spherical volume concentration fraction (svcf).

Since there is no equivalent in the literature of a two-dimensional particle distribution, we introduce the reduced volume size distribution $v_\eta(r)$, defined as the volume shape-size distribution $v(r,\eta)$ integrated over the aspect-ratio domain, i.e.

$$v_\eta(r) = \int_{\eta_{\min}}^{\eta_{\max}} v(r,\eta) \, d\eta, \tag{11}$$

where $r$ is the volume-equivalent particle radius. The function $v_\eta(r)$ is able to provide the collective trend of all contributing particle geometries in the particle distribution. However, we note that this limited (in terms of information as compared to $v(r,\eta)$) particle distribution is not directly comparable with the usual size distribution used in literature, but it can be used in order to have a general sense qualitatively, see Fig. 1 for an overview.

Furthermore, in our analysis we use data derived by inversions of sun-photometer measurements provided by the AERONET database (Holben et al. , 1998). AERONET offers a series of inversion products including the effective radius, the volume concentration, the complex refractive index, the single scattering albedo and the aerosol optical depth, which can be available in two quality levels, namely 1.5 and 2.0, for cloud screened and quality assured data respectively. Sun-photometers are passive remote sensing instruments with different operation principles from lidars and use different inversion techniques based on the theory of optimal estimation (Dubovik and King , 2000; Dubovik et al. , 2000). Although we will not expand on the latter, we will point out some incompatibilities with our approach, which are essential to recognize for our subsequent microphysical analysis. The first and most important difference is that AERONET retrievals relate to the whole atmospheric column while lidar data target specific layers and therefore the comparison cannot be quantitative. Moreover, the volume concentration size





distribution of the particle volume $u(r)$ retrieved by AERONET is defined as the derivative $\mathrm{d}u/\mathrm{d}\ln r$ with the associated total volume concentration $u_t$, both given in $\mu\mathrm{m}^3\mu\mathrm{m}^{-2}$. Following the LM-model, the volume size distribution (for lidars) is found by $v(r) = \mathrm{d}v_t/\mathrm{d}r$ and measured in $\mu\mathrm{m}^3\mu\mathrm{m}^{-1}\mathrm{cm}^{-3} = \mu\mathrm{m}^3\mu\mathrm{m}^{-2}\mathrm{m}^{-2}$.

In order to make sense of these two different measures and have some kind of comparison we turn to the quantity $\tilde{v}(\ln r) =$

$\mathrm{d}v_t/\mathrm{d}\ln r = rv(r)$, which indicates that the difference in units with $\mathrm{d}u/\mathrm{d}\ln r$ lies within a multiple of the meter. Here we apply the same concept also for our generalized model just by replacing $v(r,\eta)$ by $v_\eta(r)$. In practice, we often multiply $\tilde{v}(\ln r)$ with the aerosol layer thickness, usually several kilometers, (Osterloh et al. , 2013; Samaras et al. , 2015a). This is the motivation behind the units given in the explaining flowchart, see Fig. 1, for the so-called volume concentration size distribution (VCSD), following the terminology from Samaras et al. (2015a). Converting a reduced size distribution $v_\eta(r)$ to a VCSD is even more

involved, but we use these functions as a bridge (in terms of units) between lidar and sun-photometer. AERONET's VCSD is retrieved in 22 logarithmically equidistant bins between 0.05 and 15μm. For the shape-size distribution we use $30 \times 30$ $(r \times a)$ grid points with the radius range $[0.01, 2.2]$ (μm) and the aspect ratio range $[0.67, 1.5]$. For clarity we note that whenever we show here an one-dimensional size distribution (or VCSD) associated with the lidar, it is always implied that there was originally a shape-size distribution $v(r,\eta)$ which was first converted to it.

Second, many of AERONET's inversion products are given as a function of wavelength at least at the spectral bands of 442, 675, 872, 1020 nm where the four almucantar scans take place, in this category fall also the parameters aerosol optical depth and the refractive index (RI). Especially for the RI we will consider a spectral average of these values in order to compare with the respective parameters from our retrievals. The ranges of the real (RRI) and imaginary part (IRI) of the refractive refractive index, $1.33 \leq \mathrm{RRI} \leq 1.6$ and $0.0005 \leq \mathrm{IRI} \leq 0.5$ form the predefined grid necessary for AERONET's inversion.

Following AERONET's mode distinction in the inversion products we calculate in addition to the total effective radius, the one for the fine and coarse mode. This is done in SphInX by looking for the minimum of the shape-size distribution between $r = 0.5$ and 1 μm and setting it as the higher and the lower integration boundary for the fine and coarse mode, respectively. In contrast, here, we used a fixed separation limit $r = 0.5$μm since later we are going to compare our results with the POLIPHON method and in situ particle counter devises where essentially $r = 0.5$μm is used.

The inversion procedure does not offer a unique solution. Regularization reduces the solution space, keeping only a small number of solutions which minimize the errors. The retrieval products are compared in the following error-related quantities (ERQ), all of which are calculated as a percentage (%) using the standard deviation.

1. Variability of the solution space (Var).

2. Randomness uncertainty (Unc).

Definitions and explanations are given in the Appendix.

We extend the traditional spherical lidar setup $3\beta + 2\alpha$ which consists of 3 backscatter coefficients at 355, 532 and 1064 nm and 2 extinction coefficients at 355 and 532 nm, to $3\beta^{\parallel} + 2\alpha + (n)\beta^{\perp}$, where $\beta^{\parallel}$ and $\beta^{\perp}$ denote the horizontally and vertically polarized backscatter coefficients, respectively, and $n$ is either 1, 2 or 3 depending on the availability of Raman channels of the lidar system in 355, 532 and 1064 nm. The two cases in this paper pertain to a setup $3\beta^{\parallel} + 2\alpha + 3\beta^{\perp}$ for retrieving a shape-





and size-dependent particle distribution for the first time to the best of our knowledge. Alternatively, instead of $\beta^\perp$, the particle linear depolarization ratio $\delta = \beta^\perp/\beta^\parallel$ may appear in the setup notation to imply the same thing, and further $\alpha, \beta$ and $\delta$ may be omitted $(3+2+n)$. We note that in the case of spherical approximation, the backscatter coefficient $(\beta^\parallel)$ in the aforementioned 6-, 7- or 8-point datasets coincides with the total backscatter coefficient $\beta$.

After discretizing of the model Eq. 3 we solve the resulting linear system with regularization, which is the first step to counteract the ill-posedness of this inverse problem. SphInX is able to use different regularization techniques and parameter choice rules. The following combinations are available:

- Truncated singular value decomposition (TSVD) with the discrepancy principle,

- Tikhonov regularization with the discrepancy principle, with the L-curve method or with the generalized cross validation,

- Padé iteration with the discrepancy principle or with the L-curve method.

Details on the widely used methods TSVD and Tikhonov and the parameter choice rules can be found in most books about regularization, e.g., Hansen (2010). Padé iteration is used by the software in the context of the so-called generalized Runge-Kutta regularization methods (Böckmann and Kirsche , 2006; Böckmann and Osterloh , 2014). For our two case studies in the next section we use Padé iteration since preliminary numerical tests revealed an overall superior behavior of this iterative

regularization as compared to the other built-in methods, a fact also confirmed by further massive theoretical simulations. Extensive details on the latter as well as full functional demonstration of the software tool SphInX (a graphical-user-interface program), designed by the first author, can be found in Samaras (2017).

## 2.2   Measurement site and instrumentation

### 2.2.1   Leipzig triple-wavelength polarization Raman lidar and AERONET Sun-Photometer

We give a brief description of the triple-wavelength lidar system which provided the optical parameters. The ground-based remote sensing station was deployed at the Caribbean Institute for Meteorology and Hydrology (CIMH) in Husbands, north of the capital Bridgetown at the west coast of Barbados (13.15°N, 59.62°W, 110 m above sea level). The measurements were conducted in three SALTRACE field campaigns in June-July 2013, February-March 2014 and June-July 2014. Saharan dust plumes dominated during the summer campaigns (Haarig et al. , 2017a), whereas pure marine conditions and dust-smoke

mixtures were observed in the winter campaign (Haarig et al. , 2017b, 2019). The BERTHA lidar system is a container-based, multi-wavelength polarization Raman lidar. It has been upgraded since the SAMUM campaigns (Althausen et al. , 2000; Tesche et al. , 2011) to enable the measurement of the depolarization ratio at three wavelengths (355, 532 and 1064 nm) simultaneously, which is crucial to characterize the dust after long-range transport. A more detailed description of the 13-channel lidar system and the polarization characteristics can be found in Haarig et al. (2017a).

Currently it operates as a 3+2+3 lidar system (3 backscatter coefficients, 2 extinction coefficients and 3 depolarization ratios) with an additional water vapor channel (407 nm) and a high-spectral-resolution channel at 532 nm. In recent times, it has been used in a 3+3+2 configuration to determine the extinction at 1064 nm (Haarig et al. , 2016b). The signals are detected with a





range resolution of 7.5 m and a time resolution of 10 s. Moreover, the lidar data are smoothed with a sliding window (742.5 m) for the backscatter- and depolarization-profile and as well with 742.5 m for the calculation of the extinction-profile (range for the linear regression). Note that this suppresses the noise in the lidar data, but smooths out also the fine structure of the layers.

Additionally, an AErosol RObotic NETwork (AERONET) sun-photometer (see AERONET web page https://aeronet.gsfc.
nasa.gov/, Barbados_SALTRACE site), and a Vaisala radiosonde station (RS92 for profiling of pressure, temperature, RH, and the vector of the horizontal wind component) were operated at the field site.

### 2.2.2 Airborne Measurements

During the first SALTRACE period in June-July 2013, the ground-based measurements were complemented by airborne in situ measurements. The platform for these in situ measurements was the Falcon research aircraft of the German Aerospace
Center (DLR), which was for this purpose equipped with an extensive aerosol instrumentation. An overview of the complete instrumentation and related data products is given in Weinzierl et al. (2017) (Supplement). Information on size-resolved particle number concentrations between about 10 nm and 50 $\mu$m are acquired by a combination of wing-mounted optical particle spectrometers (OPS) and condensation particle counters (CPC) and OPS operated inside the aircraft cabin behind an isokinetic aerosol inlet. Lognormal number size distributions (in terms of volume-equivalent particle diameter) are fitted to
these data using a Bayesian inversion method (Walser et al. , 2017). Within this method, uncertainties in each instrument's response and in the particle optical properties are propagated to gain realistic size distribution uncertainties. For comparison with the results of the retrieval described in Section 2.1, the particle number size distributions derived for the inside of the dust layer are first converted into particle volume size distributions, using $\pi d_p^3/6$ for the volume of each (volume-equivalent) particle diameter $d_p$. The volume size distributions are then integrated for the sub- and supermicron diameter range to obtain
the total "fine" and "coarse" particle volume concentrations. In doing so, the initial number size distribution uncertainties are fully propagated.

## 3 Microphysical Property Retrieval

### 3.1 SALTRACE campaign case study 20 June 2014

The optical properties are thoroughly investigated in Haarig et al. (2017a). Here we give a brief overview, necessary for our
analysis. Fig. 2(a) shows the color plot of the range-corrected signal at 1064 nm (cross-polarized channel) for 20 June 2014 (case 1). We can distinguish a well defined intense aerosol layer up to 4.3 km. Later on, we will concentrate on two specific layers, namely a mixed layer containing dust and marine aerosol between 1.5 and 2.8 km and a pure dust layer between 3.3 and 4.0 km. The full overlap between the laser beam and the receiver field of view is reached at 1 km. This is the reason that several lidar profiles are only available at heights above about 1 km (more details ar given by Haarig et al. (2017a)).
Looking at the optical profiles in Fig. 2(b)-(e), the particle linear depolarization ratio at all available wavelengths is well above 20% for heights >1.5 km and reaches up to 32% ($\delta$532) at about 3.7 km, a clear indication for the presence of non-





spherical particles. Furthermore, the particle linear depolarization ratio at all available wavelengths in Fig. 2(b) shows a steady increase in the region 0.5-1.5 km from 8% to 20%. This is a clear signature of turbulence and vertical mixing of marine and dust aerosol. These meteorological processes separate the mixing layer from the dust layer according to the studies of Groß et al. (2015). We will mainly focus on the dust layer above 1.5 km height.

A reasonable layer selection is a trade-off between slowly varying lidar ratio and Ångström exponent (AE) and intense backscattering and extinction by dust particles. Focusing on the extinction-to-backscatter ratio profiles at 355 and 532 nm (not shown here, see Haarig et al. (2017a)) in the altitude range 1.5-4.3 km we have values of about $50\pm9$ sr, typical of dust presence (Ansmann et al. , 2003; Groß et al. , 2015; Heese et al. , 2009). The values of the extinction-related Ångström exponent (AE) $AE^{\alpha}$ 355/532 and of the backscatter-related Ångström exponent $AE^{\beta}$ 532/1064, see Table 2, are also characteristic of Saharan

dust particles Müller et al. (2007). Moreover, the depolarization ratio values and the lidar ratios at 355 and 532 nm fall within the ranges of aged Saharan dust found during SALTRACE campaign in 2013 (Groß et al. , 2015; Haarig et al. , 2017a). In the following, we will focus on the layers from 1.5-2.8 km and 3.3-4.0 km height. The average values and standard deviations of the $3\beta^{\parallel} + 2\alpha + 3\beta^{\perp}$ optical and intensive properties for both layers are tabulated in Table 2.

     In order to identify the aerosol source regions, the NOAA HYSPLIT model was run to compute backward trajectories

(Haarig et al. , 2017a; Draxler and Rolph , 2014), http://ready.arl.noaa.gov/HYSPLIT.php. The backward trajectories suggest air masses travelling above the African coastline (Western Sahara and Mauritania) and Mali as the dust source region. Traveling time to Barbados was about 8 days. The mean aerosol optical depth (AOD) measured with the SALTRACE AERONET sun photometer ranged from 0.37 (1640 nm) to 0.50 (340 nm) on this day.

### 3.1.1   Shape-Size Distribution and Microphysical Properties

The inversion of the optical into microphysical properties was performed with the software tool SphInX. We used Padé regularization with 100 iterations, 9-14 spline points combined with a spline degree within the range 2-5; more details on initial parameter determination can be found in Samaras et al. (2015a). Less spline points, e.g., six were ruled out because they did not behave well with the strong tendency towards radius-bimodality, which was indicated as most probable by preliminary tests. The refractive index grid (RIG) was fixed to $RRI \times IRI = [1.4, 1.5, 1.6, 1.7, 1.8] \times [0, 0.001, 0.005, 0.01, 0.05]$. The upper

integration boundary of the Fredholm equation was set to the maximum available $r_{max} = 2.2$ $\mu$m as mentioned before.

     The retrieved volume shape-size distributions are shown in 3-D in Fig. 3 and in 2-D in Fig. 4. Focusing on the altitude range 1.5-2.8 km in Fig. 4(a), we see two very well separated modes, namely a narrow fine mode and a much broader coarse mode, with maxima of 58 and 52 $\mu$m$^3\mu$m$^{-1}$cm$^{-3}$ at a radius of about 0.43 and 1.45 $\mu$m, respectively. There are contributions of all kinds of spheroidal particle geometries including spherical ones, and for higher aspect-ratios ($\eta \geq 1.20$) the peak difference

fades, so that the maxima equalize at about 30 $\mu$m$^3\mu$m$^{-1}$cm$^{-3}$. The 2-D plot in Fig. 4(a) shows a shift to higher aspect ratios for the coarse mode, i.e., to the prolate shape. Turning now to the 2-D plot for the layer 3.3-4.0 km in Fig. 4(b), we see that while the shape of the first mode is similar to the one in the lower layer, the second mode differs significantly. Large prolate particles with high aspect ratios disappear and the volume magnitude drops.



The retrieved microphysical parameters $a_t, v_t, r_{eff}$ (total, fine, coarse) $\eta_{eff}, \eta_{var}$, svcf, and RI for both layers with a step of 100 m are shown in Table 3. In this table, the variability (Var %) pertains to 5 best solutions for a single dataset corresponding to a specific altitude range, and the mean variability when more than one data sets are involved. The uncertainty (Unc %) of the mean parameter value, found for every altitude range, is also given, corresponding to different altitude ranges. Table 3 shows

that our method provides very good stability. The uncertainty of all parameters is very low and thus we could consider any of these height ranges (100 m) to arrive to quite the same result. This could be also the result of good data quality and the smoothing procedure mentioned before. It should be noted that the inversion is further aided by using the maximum number of depolarization data available to date (dataset $3\beta^{\parallel} + 2\alpha + 3\beta^{\perp}$), the benefit of which was also demonstrated in Samaras (2017) as part of the investigations for an ideal lidar setup.

Our algorithm predicts large particles. This is expected for an intense dust event. An effective radius of 0.73 $\mu$m (layer: 1.5-2.8 km) for the entire size distribution is obtained. For the fully separated modes (see Figs. , 4(a), and 5(b)), we derived an effective radius of $r_{eff}$ =0.33 $\mu$m (fine mode) and $r_{eff}$ =1.45 $\mu$m (coarse mode). Turning now to the layer from 3.3-4 km, the effective radii (total/fine/coarse) of 0.71/0.35/1.47 $\mu$m are similar to the values found for the lower dust layer (see Table 3, second panel). Another notable characteristic is that the effective radius in (3.3-4 km) is still well-retrieved despite the significant

attenuation of signal in these heights. Note, however, that the uncertainties (Unc) for the upper layer are larger, see Table 3. The lower particle concentration is already visible above 2.8 km in the lower values for the extinction and backscatter coefficients (Fig. 2). From the point of view of the microphysical retrieval it translates into a diminished surface-area- and total volume concentration (Fig. 6(d)-(i)). Fig. 6 shows the column resolved microphysical properties from 1.5 to 4.3 km. There is an overall decreasing tendency with height for the microphysical properties $v_t$ and $a_t$, but in the lower layer we see a rather steady pattern

compared to the one in the upper layer. The total surface area and volume concentration in the lower layer is larger than in the upper layer, in particular the coarse mode is threefold higher, see Figs. 5(b),6(f),(i), which, again, is in agreement with the basic signal intensity shown in Fig. 2. We want to mention finally that Tesche et al. (2019) used a completely different spheroidal model in the lidar data analysis. However the results in terms of total particle volume concentration were very similar to our findings when comparing Fig. 6(d) and Fig. 3(d) in Tesche et al. (2019).

**3.1.2   Refractive Index**

The refractive index is found to be $1.4 + 0.0497i$ for all layers between 1.5 and 4 km. Experimental findings have shown that such a high absorption can mainly be found near the sources of dust events. For instance, a soil sample from Burkina Faso (rich in hematite and kaolinite) was found in Wagner et al. (2012) using also a spheroid-particle-based inversion scheme to have $IRI = 0.0495 \pm 0.0206$ at 305 nm. Moreover, Kandler et al. (2007) at Izana (Tenerife) found that the IRI increases a lot (up

to 0.03) with decreasing particle size ($< 0.5$ $\mu$m), again due to the predominant hematite / soot component in smaller particles. In addition, Kandler et al. (2011) found at Praia (Cape Verde) between 100 and 250 nm particle diameter a mode with high absorption of around $0.1i$, which is produced by soot-sulphate mixture particles. It is also noted that the imaginary part varies depending on the source region for these dust periods. In particular, high absorbing small particles came from Mauritania, as seen also in the backward trajectories, whereas less absorbing ones came from Mali/Niger.





AERONET's retrieval suggests, on the contrary, a refractive index with RRI = 1.52 (Var: 1.02%, Unc: 1.54%) and IRI = 0.0017 (Var: 21.13%, Unc: 64.24%), i.e., weakly absorbing particles but the uncertainty is high. The given values of the complex refractive index retrieved by AERONET represent spectral mean values in the four almucantar wavelengths. AERONET observes the whole column including the predominant marine particles in the altitudes below 1.5 km which are only weakly

absorbing. Var (%) corresponds to a mean spectral variability, and the given Unc (%) corresponds to the uncertainty of the mean retrieved parameters (spectrally) between the consecutive measurements. The retrieved IRI is lower than the usual values considered for Saharan dust particles (Schladitz et al. , 2009). The latter study also finds a strong increase in IRI (up to five times higher at 637 nm) when the dust concentration is lower so that a soot-type absorber prevails. We should note that the limited resolution of the refractive index grid (Mieschka database) is an apparent shortcoming of our approach. The smoothing

effect caused by high absorption raises the degree of ill-posedness (Samaras , 2017) which at the end can lead to (under-) overestimation of the (RRI) IRI.

In this respect, Tesche et al. (2019) found that the use of depolarization input at any wavelength, i.e. 355 or 532 or 1064 nm, generally increases the retrieved values of the 532 nm SSA compared to the one obtained with the 3+2 input. The use of depolarization information leads to lower values of the imaginary part compared to the inversion in which the traditional 3+2

data set is used.

As another contribution to the discussion, Bi et al. (2018) found that a detailed shape change process from spherical particles to non-spherical particles is critical to the backscattering LDRs. To achieve high LDRs for nearly-spherical particles, the real part of the refractive index should be in the region of 1.3–1.7, and the imaginary part of the refractive index should be less than 0.01. The maximum size parameter of the high depolarization ratio depends on the imaginary part of refractive index.

### 3.1.3   Comparison with the POLIPHON Method and AERONET retrievals

The fine and coarse mode fraction of the volume concentration derived with our method was compared with the products obtained with the POLIPHON separation method Mamouri and Ansmann (2017) and with respective AERONET results. The comparison is shown in Fig. 7. With respect to the retrieval of the particle size distribution the spheroidal particle approximation used in our inversion approach is now well attested as the superior method compared to the Mie-scattering-based method.

The comparison with the POLIPHON shows reasonable agreement. This is a remarkable finding when keeping in mind that spheroids by no means capture the true morphological nuances of dust and that the inversion is an ill-posed problem, i.e., small deviations on input may produce high deviations on output. The graphs in Fig. 7 show the same qualitative behaviour through the whole range for both modes. The average absolute difference in volume concentration (POLIPHON/AERONET vs inversion method) with respect to fine mode fraction is about 5.5 $\mu$m$^3$cm$^{-3}$ only. Note that for the coarse mode fraction

the inversion makes a cutoff at $r = 2.2$ $\mu$m, which already explains the found differences to the POLIPHON and AERONET results.





## 3.2 SALTRACE campaign case study 10 July 2013

A strong and long-lasting Saharan dust outbreak reached Barbados between 9 and 13 July 2013, substantially stronger than in June 2014 (case 1). On 10 July 2013, the BERTHA lidar system and the Falcon aircraft measured the dust plume which reached almost 5 km in height (Fig. 8). More specifically, this intense Saharan dust layer extended from 2.0-4.5 km height and

showed a 532 nm particle depolarization ratio of $0.27\pm0.015$ (Fig. 8, Table 4), and a lidar ratio of $51\pm3$ sr. Downward mixing of mineral dust to the marine aerosol layer was observed below 2 km height. The evening AOD was again high with values from 0.35 (1640 nm) to 0.50 (340 nm) according to the AERONET observations.

In the following, the lidar measurements from 10-11 July 2013 between 23:15 and 00:45 UTC are investigated. Local sunset was at 22:29 UTC. We focus on the height range from 2-3 km. The dust optical properties are summarized in Table 4.

### 3.2.1 Microphysical Properties and Comparison with Case 1

It is not surprising that the shape-size distributions in Fig. 9 and Fig. 10 are similar to the ones for case 1. The HYSPLIT backward trajectories showed similar long-range transport features (Haarig et al. , 2017a). SphInX produces again stable retrievals in the same sense as presented earlier, i.e, running twice for the same layer considering the range from 2-3 km (i) as a whole (single run) and (ii) breaking it into ranges of 100 m (multiple runs). The results are summarized in Table 5. There

is virtually no difference between the retrieved values for whole range and the mean retrieved values for the split range and with very low uncertainties. For the total volume concentration we get 57.23 and 56.10 $\mu m^3 cm^{-3}$, respectively. Fig. 12 shows the microphysical properties for the whole aerosol column from 1.5-5.0 km. Within the considered layer 2-3 km the volume concentration shows its largest value (Fig. 12(d)-(f)). The partly strong variability in the retrievals results and peaking structures in the profiles have to be interpreted with caution and are attributed to the sensitive impact of small uncertainties in the optical

properties on the inversion products.

Fig. 11 shows the VCSD for three different layers. The decrease of the coarse mode particle concentration in the layer from 3-4.5 km (blue line) when compared to the respective coarse mode number concentration in the lower layer from 2-3 km (orange line) is in line with the reduced backscatter and extinction strength (Fig. 8). Focusing on the whole layer from 1.5-5 km height (pink lines in Fig. 5(b) for case 1, Fig. 11 for case 2), the maximum values for fine and coarse mode are 8 and

12 $\mu m^3 cm^{-3}$ (case 1) and 29 and 44 $\mu m^3 cm^{-3}$ (case 2). The values for case 2 are again larger since the dust event was more intense. All this is in agreement with the results shown in Fig. 2 and 8 and presented in the Table 2 and 4.

When comparing the retrieval parameters in Tables 3 and 5, we find that the total effective radii are very close to each other in all three layers, namely, 0.73, 0.71 and 0.78 $\mu m$ and that the parameters svcf, $\eta_{eff}$ and $\eta_{var}$ are equal. This points to the fact that the aged dust plumes originated from similar dust sources. On the other hand, the values for total surface-area (174.09,

70.32 and 215.61 $\mu m^2 cm^{-3}$) and volume concentration (42.46, 16.51 and 56.10 $\mu m^3 cm^{-3}$) differ significantly. They are larger for case 2, 10 July 2013, reflecting the stronger dust outbreak measured in July 2013.



### 3.2.2 Comparison with POLIPHON method

The retrieved fine and coarse mode volume concentration were compared with the respective POLIPHON results. The graphs in Fig. 13 show almost the same qualitative behaviour (monotonicity) throughout the entire height range for both modes and agree quite well. We should mention again that the coarse mode fraction derived by the POLIPHON method (thick green line)

is not limited by the cutoff ($r = 2.2~\mu$m) and is therefore larger. The absolute difference in volume concentration between POLIPHON method and inversion with respect to fine mode fraction is, on average, $7.81~\mu\mathrm{m}^3\mathrm{cm}^{-3}$.

### 3.2.3 Comparison with FALCON Measurements

The Falcon aircraft performed measurements on 10 July 2013 Haarig et al. (2019). We consider the observations taken at height levels of 2594, 3560, 4204, and 4369 m during the time periods from 16:46-16:55, 18:12-18:21, 17:52-18:10, and 16:30-16:40

UTC, respectively. The horizontal distances to the lidar site were about $220\pm2, 20\pm7, 66\pm45$, and $220\pm2$ km. A comparison of the Falcon in situ measurements of fine and coarse mode mass concentration to the coincident daytime lidar measurements (17:01–19:25 UTC) shows good agreement (Haarig et al. , 2019). The dust optical properties did not change between lidar measurements of day- and nighttime.

Here we compare the inversion retrieval results with airborne measurements of particle volume concentrations. The FAL-

CON observations are included in Fig. 13. By comparing the coarse mode volume concentrations, we found good agreement. The mean relative difference is 39 %, and 14 % if we compare the inversion results with the particle counter 84-percentile value. The inversion algorithm overestimates the fine mode volume concentration. However, taking into account all differences, e.g., time and horizontal distances between the in situ and lidar measurements and measurement errors in the lidar and particle counter data the agreement is good.

### 3.3 Comparison with AERONET Sun-photometer

We compare the volume concentration size distributions found by lidar-based- and AERONET retrievals. The flowchart in Fig. 1 provides an overview of the different retrieval procedures and products. Fig. 5(b) and Fig. 11(b) display the lidar-based volume concentration size distribution for three layers and Fig. 5(a) and Fig. 11(a) show the ones retrieved by the AERONET data analysis method. The latter figures show merely a very prominent coarse mode predicting very large particles.

The complete absence of a fine mode is often a mathematical artifact when one of the expected modes is much more dominant than the other. The smaller one is either suppressed or smoothed out. Although the lidar-based- and AERONET VCSDs cannot be directly compared, we can see that the volume concentration values from the two derivations (lidar and AERONET) are within the same order of magnitude for the layers 1.5-4.3 km (case 1, Fig. 5a) and 1.5-5.0 (case 2, Fig. 11a). Thus, considering the thickness of the whole dust layer of about 4 and 4.5 km, respectively, the estimated maximum lidar-based column volume

concentration is about $0.12~\mu\mathrm{m}^3\mu\mathrm{m}^{-2}$ and $0.20~\mu\mathrm{m}^3\mu\mathrm{m}^{-2}$ which is comparable with the respective AERONET maximum values of about $0.16~\mu\mathrm{m}^3\mu\mathrm{m}^{-2}$ and $0.19~\mu\mathrm{m}^3\mu\mathrm{m}^{-2}$. Because the dust volume concentration was not height-independent within the dust layer from 1.5 to 4-5 km height and there were additional contributions from the layer below 1.5 km in both cases,



the comparison of profile and column-integrated observations remains difficult. Nevertheless, the absolute differences are very small in particular in case 2, and the maximum peaks in Fig. 5 (a) and (b) and Fig. 11(a) and (b) occur at similar radius values, namely at 1.71 and 1.75 $\mu$m.

Finally, we included the column-integrated volume concentrations from the sun-photometer, separated into fine and coarse mode, in Figs. 7 and 13. Evidently, the qualitative comparison between the sun-photometer retrieval and the POLIPHON method shows good agreement for the fine modes for both cases and a larger deviation for the coarse modes, especially for case 2.

## 4   Conclusions

In this study, we derived the microphysical properties for two dust events of different intensity by using input data from a
Raman lidar system with 3 depolarization channels (355, 532, 1064 nm). The measurements were performed at Barbados during the SALTRACE campaign in 2013 and 2014. Furthermore we conducted a series of comparisons with independent retrievals with the POLIPHON method, AERONET and in situ airborne measurements.

We followed a natural generalization of the Mie model in two dimensions by considering spheroidal optical efficiencies which refines the traditional size distribution to a shape-size distribution (radius and aspect-ratio dependent) and introduces
new shape-related parameters. The inherently unstable nature of the retrieval calls for a careful selection of layers so that the aerosol type (characterized by lidar ratio, AE) is reasonably confined, potentially turbulent parts are left out, and at the same time, the most intense parts of the optical profiles are included. This gave the opportunity to focus on the stability of the algorithm and its response to particle size differences and signal attenuation. By performing retrievals to subsequent sub-layers (100 m) within the entire dust layer we showed that the algorithm is able to sense the variation of particle size for case 1 as
predicted previously by looking at AE profiles. We also saw that as long as the intensive properties are relatively constant the microphysical properties calculated either within a layer as a whole or within smaller sub-layers have tenuous differences. A good quality of the optical data and adequate smoothing procedure obviously play a role as well.

Equally large particles were found for both cases ($r_{\mathrm{eff}}^{\mathrm{mean}} = 0.75$ $\mu$m), but case 2 was a more intensive dust event with 24% larger surface-area and volume concentrations. The shape parameters svcf, $\eta_{\mathrm{eff}}$ and $\eta_{\mathrm{var}}$ were identical which is an indication
for similar origin of the aged dust particles. The shape-size distribution revealed two distinct modes with spherical and non-spherical contributions for both aged dust cases. The coarse mode was clearly prevailing as expected for dust-like particles. The retrieved IRI (0.05) indicate high absorption, which may occur under particular circumstances we briefly explored. In marked contrast, AERONET produces a possible underestimated IRI (0.0017).

A rough comparison for the retrieved volume concentrations between AERONET and our inversion could be done by sum-
ming up all shape contributions of a shape-size distribution to produce an analogue of a (1-D) size distribution and then scaling it to the corresponding columnar range. We found that for both cases the orders of magnitude match, and notably there was even some quantitative resemblance in the volume concentration values (low absolute differences) and the radii of the maximum points of the coarse modes, all of which, again, are merely of qualitative interest. A comparison between the POLIPHON





method and our inversion revealed remarkable similarities in the trend of the volume concentration with respect to height, with an average absolute difference of 6.6 $\mu m^3 cm^{-3}$ (mean for both cases). In the same coarse-grained sense and taking into account the essential temporal and procedural differences, the in airborne situ measurements, available only for case 2, were consistent with the retrieved volume concentration, showing an average (84% percentile) deviation of 14 %.

In conclusion, the microphysical retrieval results derived with our approach provide an adequate description of the aged dust cases, and the multiple comparisons with different and independently obtained data analysis and measurement approaches corroborate the strength of the inversion algorithm. As stressed throughout this paper, the substantial raise in complexity by considering the spheroidal approximation and the difficulty in retaining numerical stability forces the use of offline calculations (kernel database and discretizations). This in turn restricts significantly the solution framework in particle shape (aspect ratio),

size (radius) and chemical composition (RI). A well-grounded and argued extension of the software tool for the future concerns the considered size range. The current radius cutoff seems to underestimate the retrieved parameters and also force the shape-size distribution to have a steeper tail. Although all known approaches have several restrictions on chemical composition via a predefined refractive index grid and particle shape (e.g. use of experimental aspect ratio distributions), it is well-known that this practice has severe effects on the retrieval outcome and therefore these ranges have to be extended as well. For instance, the

oscillatory trend of backscattering efficiencies fades away (smoothing effect) when particles become ever more non-spherical or absorbing (Samaras , 2017) and therefore a greater resolution and diversity in RI and aspect ratios may facilitate the inversion. Finally, it also remains open for future investigations whether it is necessary to extend the retrieval database with respect to a wavelength-dependent refractive index.

**Appendix A**

Here, we give definitions and explanations of ERQ reported in Section 2.1 (Samaras , 2017):

1.  Variability of the solution space (Var). The algorithm is completed by ordering the solutions with respect to increasing error level and choosing a few of the first ones (calculate the mean solution out of a few least-residual solutions). This uncertainty percentage is the standard deviation of a sought parameter, which is derived by the chosen best solutions, divided by the mean value of the parameter. It describes how much a mean value of a parameter varies from all best

solutions in the same solution space i.e. for a specifier error level $\epsilon$. In this regard there are two interpretations of such a statistical measure, both of which have to do with potential solution clusters. We rely on these clusters to find physically meaningful solutions, since previous experience with the sphere-particle approximation, showed that most of the mathematically acceptable solutions do not qualify physically. Therefore, on the one hand, the solution space should be variable enough in its full extent (all solutions), i.e. to produce clusters of more physically probable solutions. On the

other hand, there should be a relative homogeneity for a small sample of "best" solutions (small residual error), reflecting the ability of the examined method to recognize such clusters. More details on clusters and patterns in solutions spaces for real-data inversions can be found in Samaras et al. (2015a). For more than one data-sets, Var represents the mean variability of all produced solution spaces.

2. Randomness uncertainty (Unc). In simulation studies this is related to the stability of the examined method with respect to several repetitions of a numerical experiment of the same simulated atmospheric scenario but with different (random) instances of the same error level. By extension, this obviously characterizes the capacity of the method to reproduce well a possibly accurate result. The value of Unc is derived by first calculating the mean value of a parameter for every data-set of different data error, and then divide the standard deviation of these values by their mean. In other words, Unc is a form of Var with respect to the different-error data-sets. For measurement cases, these data-sets could consist of optical data values related to consecutive smaller "sub-layers" of a layer which is partitioned in order to keep intensive parameters (e.g. AE, LR) relatively constant, and therefore Unc still makes sense as an additional measure of variability among the retrieved solutions.

*Author contributions.* SS investigated the topic and results under the supervision of CB. SS implemented the software tool SphInX and made the microphysical retrievals from lidar data. CB investigated the comparisons with the polarization lidar-photometer networking method, ground-based photometer and airborne in situ particle counter data with the helpful comments and discussions of SS, MH, AA, AW and BW. CB and SS prepared the manuscript. AA, BW, MH and AW read and supported the discussion and provided comments on the manuscript. MH performed the lidar measurements at Barbados and calculated the optical properties and gave the layer information as input in the inversion calculation. BW and AW provided the in situ concentration data.

*Competing interests.* The authors declare that they have no conflict of interest.

*Acknowledgements.* We would like to thank the reviewers for their thoughtful comments and efforts towards improving our manuscript.

The research of the first two authors leading to these results has received funding mainly from the European Union Seventh Framework Programme for research, technological development and demonstration under grant agreement No. 289923 – ITaRS. Moreover, they are thankful to Tom Rother for providing the software Mieschka which was vital for the construction of the discretization database. Moreover, this activity is supported by ACTRIS Research Infrastructure (EU H2020-R/I) under grant agreement number 654109. BW and AW have received funding from the European Research Council (ERC) under the European Union's Horizon 2020 research and innovation programme under grant agreement No. 640458 (A-LIFE). The SALTRACE aircraft field experiment was funded by the Helmholtz Association (Helmholtz-Hochschul-Nachwuchsgruppe AerCARE, grant agreement no. VH-NG-606) and by DLR.

The lidar observations (level 0 data and measured signals) and photometer data as well as the analysis products are available at TROPOS upon request (info@tropos.de). In situ data are available on request to bernadett.weinzierl@univie.ac.at.





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

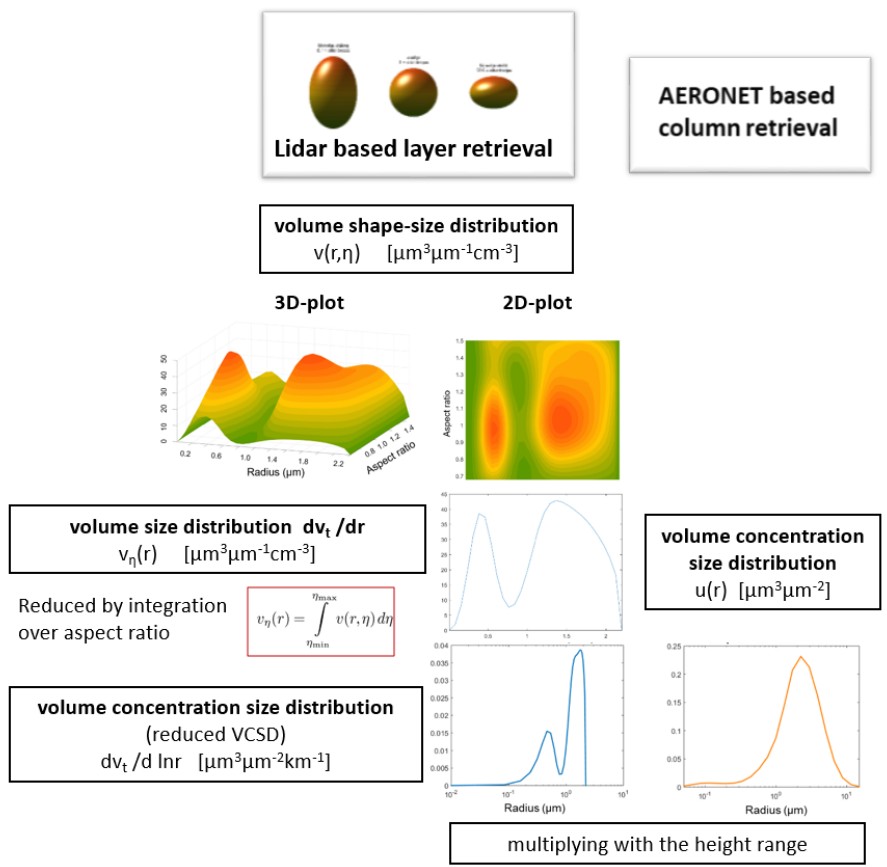

**Figure 1.** Qualitative flowchart (top to bottom) illustrating the transformations made in order to compare the orders of magnitude of the volume concentration between the lidar-based retrieval (thick blue single line) and the AERONET sun-photometer retrieval (thick orange single line): A two-dimensional particle distribution is retrieved as a function of the radius and the aspect ratio using lidar data from a specified layer. By integrating over all aspect-ratio contributions we end up with an one-dimensional size distribution (reduced VCSD). The difference in units of the latter and the AERONET-based size distribution motivates the final step, i.e. taking the product of the reduced VCSD with the particular column height.

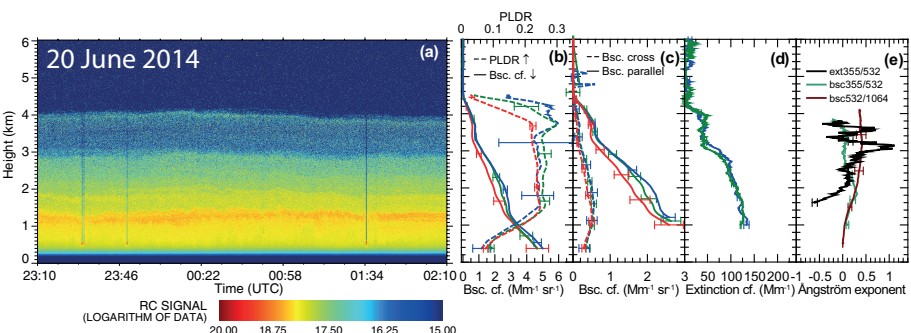

**Figure 2.** (a) Height-time display of the range-corrected lidar signal at 1064 nm (cross-polarized signal) for case 1 (20 June 2014), collected at the SALTRACE lidar station in Barbados. An intense dust layer is detected, reaching 4.3 km a.g.l. (b)-(e) Profiles of the retrieved optical properties from left to right: (b) total particle backscatter coefficient at 355 (blue), 532 (green) and 1064 nm (red line), particle linear depolarization ratio (PLDR) at 355, 532 and 1064 nm (dashed lines in the same colours), (c) backscatter coefficiens compited from the cross and parallel-polarized signal profiles, (d) extinction coefficient at 355 and 532 nm as well as (e) three Ångström coefficients. A vertical smoothing window of 99 bins (742m) was applied to the profiles in (b)-(e).

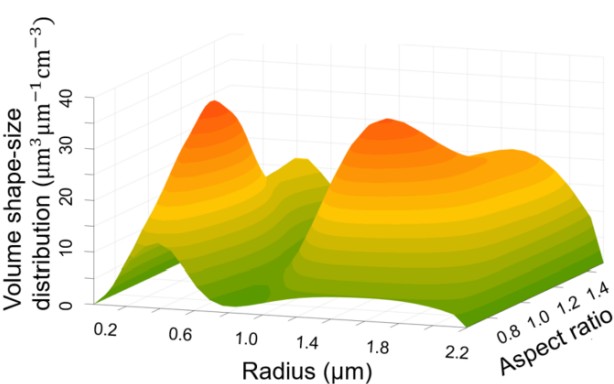

**Figure 3.** The retrieved shape-size distribution (average of single 100 m retrievals) shown in 3-D for case 1 (20 June 2014) for the entire dust layer from 1.5-4.3 km height.

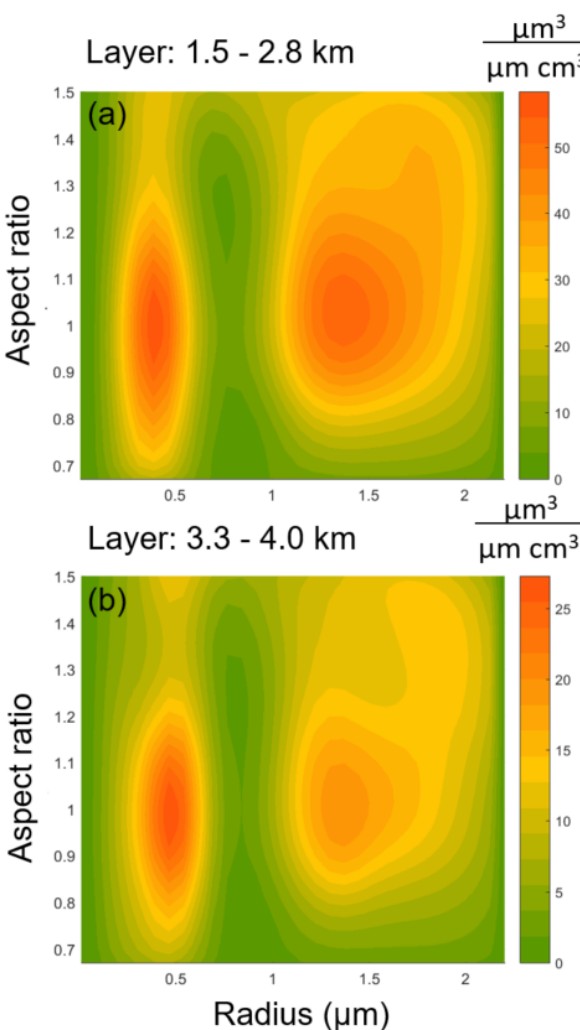

**Figure 4.** The retrieved shape-size distribution (average of single 100 m retrievals) shown in 2-D for case 1 (20 June 2014), in the upper panel for the layer from 1.5-2.8 km and in the lower panel for the layer from 3.3-4.0 km.

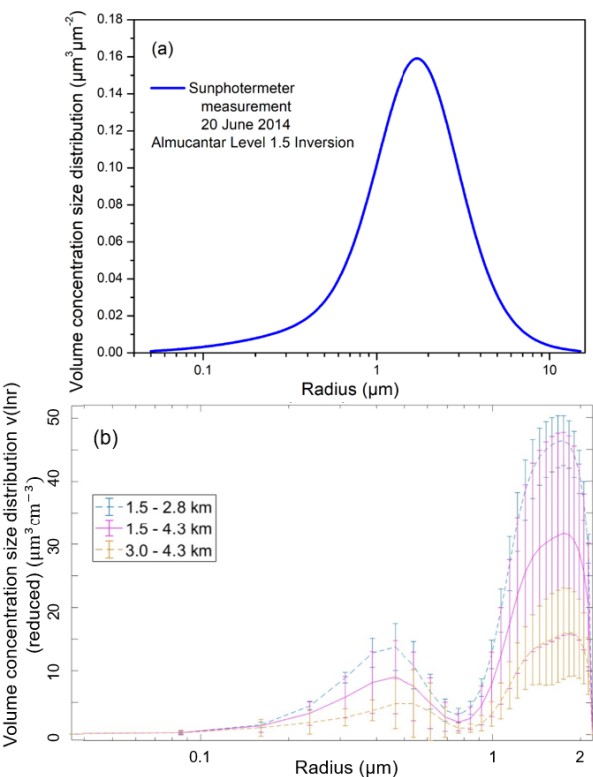

**Figure 5.** Upper panel: AERONET's VCSD for 20 June 2014 (case 1). The sun-photometer-based distribution corresponds to the mean retrieved distribution from consecutive measurements in the evening (AERONET). Lower panel: The retrieved reduced volume concentration size distribution (average of single 100 m retrievals) for the layer from 1.5-2.8 km (blue), the layer from 3-4.3 km (orange) and for the entire layer from 1.5-4.3 km height (pink). The uncertainty bars belong to the uncertainties with respect to the different layer bins (100 m) within the specified altitude range.



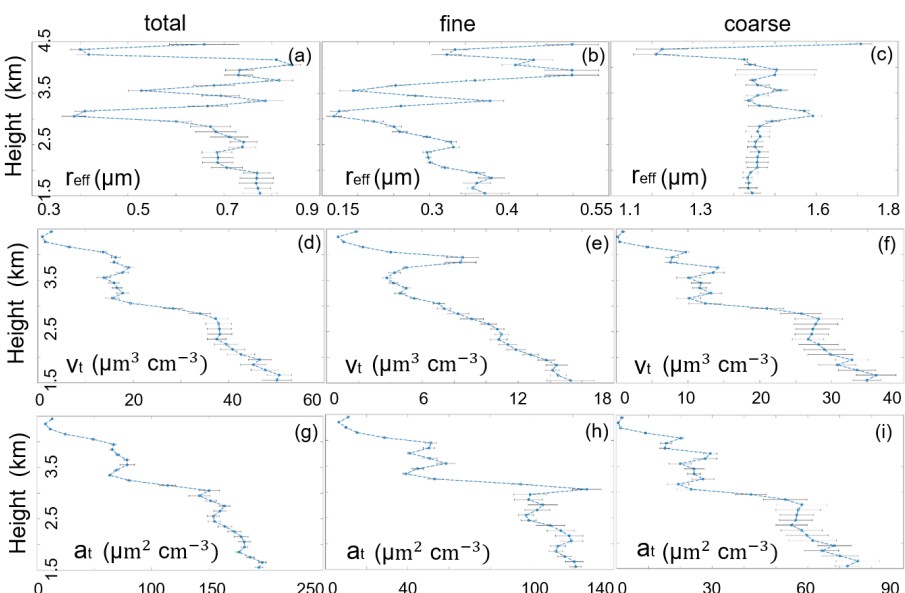

**Figure 6.** Microphysical properties: effective radius $r_{\mathrm{eff}}$, total volume $v_{\mathrm{t}}$ and surface-area concentration $a_{\mathrm{t}}$ from lidar retrieval for case 1 (20 June 2014), for fine, coarse and total mode. The variability bars are calculated from the chosen (5) least-residual solutions for each 100 m retrieval of the lidar-based inversion.



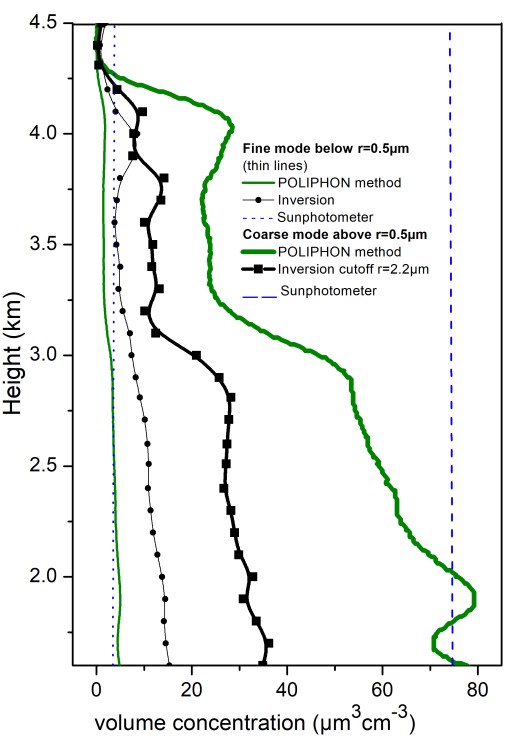

**Figure 7.** Comparison of total volume concentration $v_t$ separated into fine and coarse mode fractions obtained with the SphInX inversion and, alternatively, with the POLIPHON method and the AERONET data analysis scheme. Comparisons are presented for case 1 (20 June 2014). Note, that the coarse mode in the SphInX inversion data analysis extends up to 2.2 $\mu$m, only.



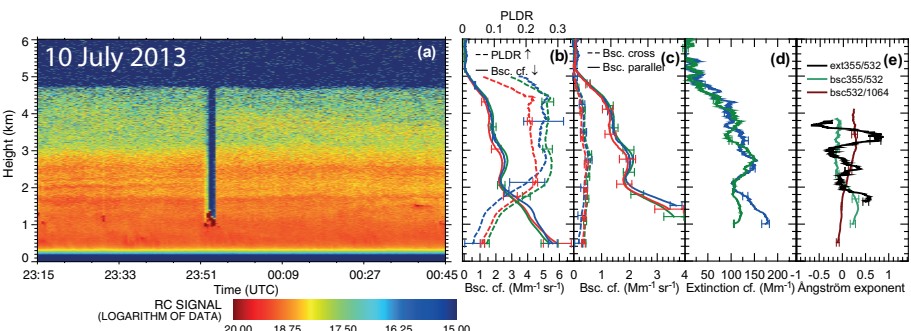

**Figure 8.** Same as Fig. 2 but for case 2, 10 July 2013. (a) (a) Height-time display of the range-corrected lidar signal at 1064 nm (cross-polarized signal) collected at the SALTRACE lidar station in Barbados. An intense dust layer is detected up to 4.8 km a.g.l. (b) -(e) Same as Fig. 2 but for case 2 (10 July 2013).



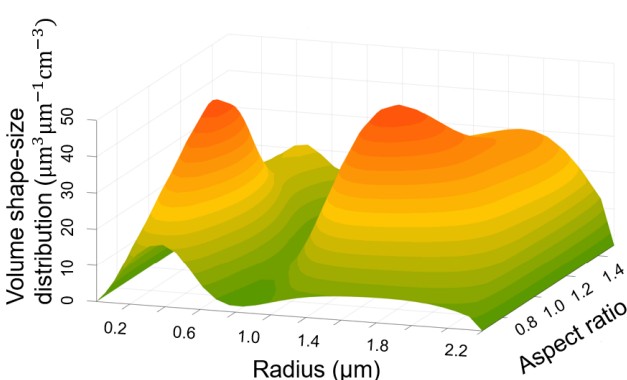

**Figure 9.** The retrieved shape-size distribution (average of single 100 m retrievals) shown in 3-D for the layer from 1.5-5 km height for case 2 (10 July 2013).

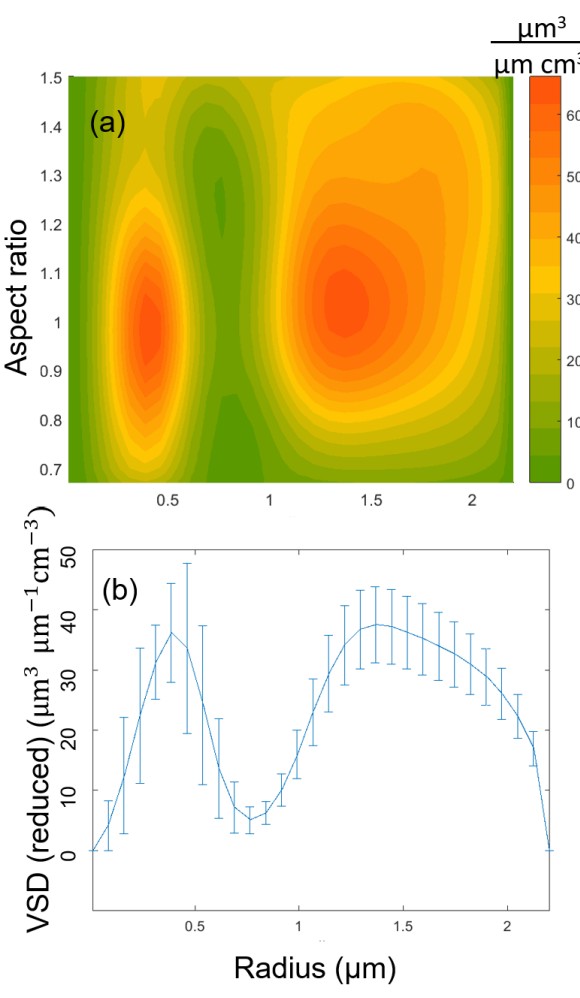

**Figure 10.** Upper panel: The retrieved shape-size distribution (average of single 100 m retrievals) for the dust layer 2-3 km shown in 2-D for case 2 (10 July 2013). Lower panel: Corresponding volume size distribution (reduced) with uncertainty bars belonging to the uncertainties with respect to the different layer bins (100 m) within the specified altitude range.

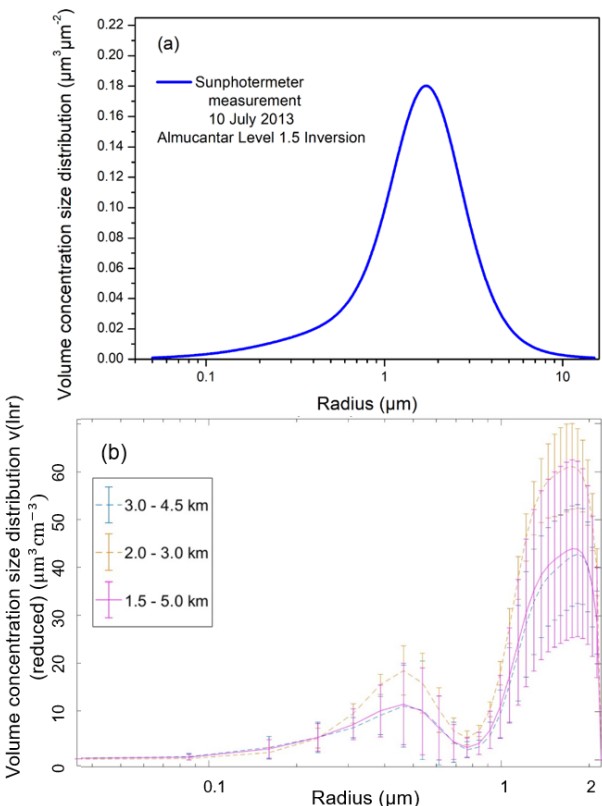

**Figure 11.** Upper panel: AERONET's VCSD for 10 June 2013 (case 2). The sun-photometer-based distribution corresponds to the mean retrieved distribution from consecutive measurements in the evening (AERONET). Lower panel: The retrieved reduced volume concentration size distribution (average of single 100 m retrievals) for the layer from 3-4.5 km (blue), the layer from 2-3 km (orange) and the entire layer from 1.5-5 km height (pink). The uncertainty bars belong to the uncertainties with respect to the different layer bins (100 m) within the specified altitude range.





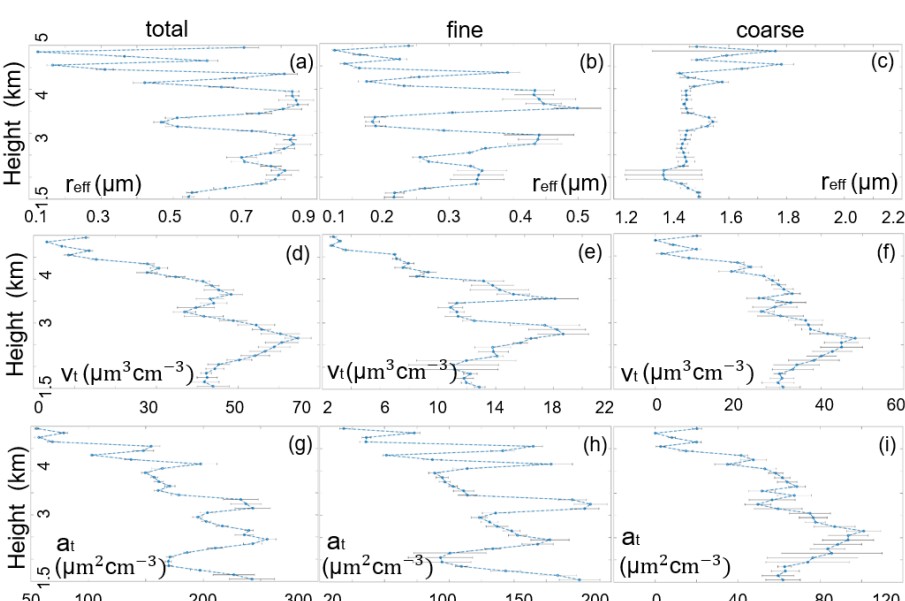

**Figure 12.** Same as Fig. 6 but for case 2 (10 July 2013).

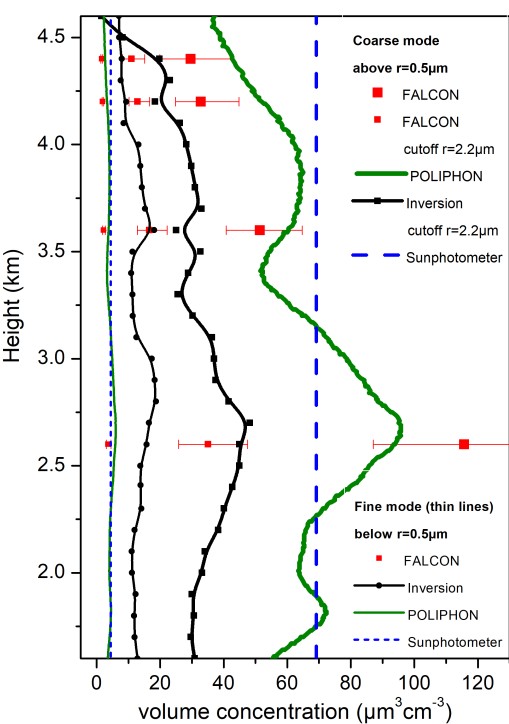

**Figure 13.** Case 2 (10 July 2013): Comparison of total volume concentration $v_t$ separated into fine and coarse mode fractions obtained by applying the SphInX inversion and POLIPHON method and derived from airborne FALCON in situ measurements at four different flight levels. With respect to the later ones in each height and mode the median values (square symbol) as well as the percentiles 16 (minus error bar) and 84 (plus error bar) are given. Additionally, the column-integrated values derived from the AERONET sun-photometer observations are included.





**Table 1.** Parameters used from the database of Mieschka software (Rother and Kahnert , 2014). $\Re(m)$, $\Im(m)$ are real and imaginary part of refractive index.

| $\Re(m)$ | 1.33 | 1.4 | 1.5 | 1.6 | 1.7 | 1.8 | |
|---|---|---|---|---|---|---|---|
| $\Im(m)$ | 0 | 0.001 | 0.005 | 0.01 | 0.03 | 0.05 | 0.1 |
| aspect ratio | 0.67 | 0.77 | 0.87 | 1 | 1.15 | 1.3 | 1.5 |
| scattering quantities | extinction efficiency | | backscatter efficiency $Q_\beta^{\parallel}$ | | backscatter efficiency $Q_\beta^{\perp}$ | | SSA |





**Table 2.** Optical and intensive properties for case 1 (20 June 2014) derived by Barbados lidar station, averaged in the dust layers 1.5-2.8 km and 3.3-4.0 km, see Fig. 2. ($\beta$, $\alpha$, $\delta$ are backscatter-, extinction coefficient and depolarization ratio, LR, AE are lidar ratio and Ångström exponent.)

| Optical properties | | | | |
|---|---|---|---|---|
| $\alpha 355$ (Mm$^{-1}$) | $\beta 355^{\parallel/\perp}$ (Mm$^{-1}$sr$^{-1}$) | $\alpha 532$ (Mm$^{-1}$) | $\beta 532^{\parallel/\perp}$(Mm$^{-1}$sr$^{-1}$) | $\beta 1064^{\parallel/\perp}$(Mm$^{-1}$sr$^{-1}$) |
| Layer: | 1.5-2.8 km | | | |
| 102 | 1.90 / 0.46 | 105 | 1.74 / 0.45 | 1.45 / 0.35 |
| ±10 | ±0.35 / ±0.08 | ±13 | ±0.28 / ±0.07 | ±0.29 / ±0.07 |
| Layer: | 3.3-4.0 km | | | |
| 40 | 0.55 / 0.15 | 42 | 0.56 / 0.15 | 0.46 / 0.10 |
| ±4 | ±0.09 / ±0.02 | ± 5 | ±0.06 / ±0.02 | ±0.04 / ±0.02 |
| Intensive properties | | | | |
| $\delta 355/532/1064$ (%) | LR355 (sr) | LR532 (sr) | AE$^{\alpha}$ 355/532 | AE$^{\beta}$ 532/1064 |
| Layer: | 1.5-2.8 km | | | |
| 24.2 ± 5.7/25.9 ± 1.7/23.9 ± 0.8 | 46 ± 3 | 48 ± 5 | 0.05 ± 0.01 | 0.29 ± 0.08 |
| Layer: | 3.3-4.0 km | | | |
| 26.8 ± 14.8/27.2 ± 2.9/21.1 ± 1.0 | 59 ± 6 | 59 ± 9 | −0.05 ± 0.10 | 0.37 ± 0.10 |





**Table 3.** Retrieved microphysical parameters for the two dust layers from 20 June 2014 (case 1) captured by Barbados's lidar station. The variability (Var %) is calculated from the chosen (5) least-residual solutions for the lidar-based inversion. The uncertainty (Unc %) corresponds to the parameter uncertainty with respect to the different layer bins (100 m) within the specified altitude range. ($a_t$, $v_t$, $r_{eff}$ are total surface area - and volume concentration and effective radius, $\eta_{eff}$, $\eta_{var}$, svcf are volume-weighted-effective aspect radio, aspect ratio width and spherical volume concentration fraction.)

Lidar-based inversion: $\mathrm{RRI} = 1.4$, $\mathrm{IRI} = 0.05$

| Parameter | $a_t$ (total/fine/coarse) | $v_t$ (total/fine/coarse) | $r_{eff}$ (total/fine/coarse) | svcf | $\eta_{eff}$ | $\eta_{var}$ |
|---|---|---|---|---|---|---|
| Unit | $\mu m^2 cm^{-3}$ | $\mu m^3 cm^{-3}$ | $\mu m$ | — | — | — |

1.5- 2.8 km, step: 100 m

| | $a_t$ | $v_t$ | $r_{eff}$ | svcf | $\eta_{eff}$ | $\eta_{var}$ |
|---|---|---|---|---|---|---|
| Average | 174.09/ 111.46/ 62.54 | 42.46 / 12.29 / 30.18 | 0.73 / 0.33 / 1.45 | 0.31 | 1.10 | 0.046 |
| Variability % | 2.71 / 4.44 / 9.68 | 6.17 / 4.93 / 8.82 | 4.51 / 3.14 / 1.41 | 4.13 | 0.52 | 2.90 |
| Uncertainty% | 8.06 / 7.36 / 11.10 | 11.72 /16.04 /10.49 | 5.16 /11.81 /0.74 | 1.56 | 0.21 | 1.14 |

3.3- 4.0 km, step: 100 m

| | $a_t$ | $v_t$ | $r_{eff}$ | svcf | $\eta_{eff}$ | $\eta_{var}$ |
|---|---|---|---|---|---|---|
| Average | 70.32 / 47.92 / 22.36 | 16.51 / 5.58 / 10.93 | 0.71 / 0.35 / 1.47 | 0.32 | 1.10 | 0.046 |
| Variability % | 3.57/ 5.37/ 11.70 | 6.63/ 7.50/ 10.95 | 4.72/ 5.90/ 2.74 | 3.96 | 0.90 | 4.31 |
| Uncertainty% | 8.60/ 14.17/ 25.23 | 9.90/ 36.05/ 23.69 | 13.15/ 33.56/ 2.15 | 5.96 | 0.59 | 1.87 |



**Table 4.** Dust optical properties observed with lidar in the dust layer from 2-3 km height above Barbados. $\beta$, $\alpha$, $\delta$ are backscatter-, extinction coefficient and depolarization ratio, respectively. LR, AE are lidar ratio and Ångström exponent, respectively.

| Optical properties | | | | |
|---|---|---|---|---|
| $\alpha355$ (Mm$^{-1}$) | $\beta355^{\|\|/\perp}$ (Mm$^{-1}$sr$^{-1}$) | $\alpha532$ (Mm$^{-1}$) | $\beta532^{\|\|/\perp}$ (Mm$^{-1}$sr$^{-1}$) | $\beta1064^{\|\|/\perp}$ (Mm$^{-1}$sr$^{-1}$) |
| 130 | 1.94 / 0.48 | 130 | 2.00 / 0.55 | 1.84 / 0.42 |
| $\pm16$ | $\pm0.07$ / $\pm0.05$ | $\pm14$ | $\pm0.11$ / $\pm0.04$ | $\pm0.10$ / $\pm0.03$ |
| Intensive properties | | | | |
| $\delta355/532/1064$ (%) | LR355 (sr) | LR532 (sr) | AE$^\alpha$ 355/532 | AE$^\beta$ 532/1064 |
| $24.9\pm5.2/27.7\pm1.5/22.9\pm0.8$ | $53\pm5$ | $51\pm3$ | $-0.04\pm0.18$ | $0.17\pm0.06$ |





**Table 5.** Retrieved microphysical parameters for the dust layer from 10 July 2013 (case 2) captured by Barbados's lidar station. The variability (Var %) is calculated from the chosen (5) least-residual solutions for the lidar-based inversion. The uncertainty (Unc %) corresponds to the parameter uncertainty with respect to the different layer bins (100 m) within the specified altitude range.($a_\mathrm{t}$, $v_\mathrm{t}$, $r_\mathrm{eff}$ are total surface area - and volume concentration and effective radius, $\eta_\mathrm{eff}$, $\eta_\mathrm{var}$, svcf are volume-weighted-effective aspect radio, aspect ratio width and spherical volume concentration fraction.)

Lidar-based inversion: $\mathrm{RRI} = 1.4$, $\mathrm{IRI} = 0.05$

| Parameter | $a_\mathrm{t}$ (total) | $v_\mathrm{t}$ (total) | $r_\mathrm{eff}$ (total) | svcf | $\eta_\mathrm{eff}$ | $\eta_\mathrm{var}$ |
|---|---|---|---|---|---|---|
| Unit | $\mu\mathrm{m}^2\mathrm{cm}^{-3}$ | $\mu\mathrm{m}^3\mathrm{cm}^{-3}$ | $\mu\mathrm{m}$ | — | — | — |

2.0 - 3.0 km

| | $a_\mathrm{t}$ | $v_\mathrm{t}$ | $r_\mathrm{eff}$ | svcf | $\eta_\mathrm{eff}$ | $\eta_\mathrm{var}$ |
|---|---|---|---|---|---|---|
| Average | 213.00 | 57.23 | 0.71 | 0.32 | 1.10 | 0.045 |
| Variability % | 2.19 | 5.37 | 3.36 | 3.40 | 0.56 | 2.18 |

2.0 - 3.0 km, step: 100 m

| | $a_\mathrm{t}$ | $v_\mathrm{t}$ | $r_\mathrm{eff}$ | svcf | $\eta_\mathrm{eff}$ | $\eta_\mathrm{var}$ |
|---|---|---|---|---|---|---|
| Average | 215.61 | 56.10 | 0.78 | 0.31 | 1.10 | 0.045 |
| Variability % | 2.47 | 6.34 | 4.55 | 4.67 | 0.55 | 3.02 |
| Uncertainty% | 12.77 | 9.90 | 6.70 | 2.07 | 0.33 | 1.10 |