# Peer review of "Retrieval of microphysical dust particle properties from SALTRACE lidar observations: Case studies"

_Atmospheric Chemistry and Physics, 2020_

## Referee Comment (RC1) · Anonymous Referee #3 · 6 Jul 2020

The manuscript presents the characterization of two dust events during the SALTRACE campaign, using the SphInX retrieval algorithm. The manuscript should go under major revisions, based on the suggestions in my initial manuscript evaluation, which are explained in more detail herein (the initial manuscript evaluation is provided at the end for completeness). Based on these suggestions, this work is scientifically significant not for the characterization of dust during SALTRACE (which is unsuccessful in my opinion), but for showing that SphInX retrieval algorithm is quite limited in characterizing dust particles, at least in its current version. Future versions may facilitate its usage for dust particles, since the authors mention the extension of the algorithm to larger sizes and different aspect ratios, which is vital for the proper retrieval of dust, as discussed

below. Another suggestion is that since this paper should present the limitations of SphInX in retrieving dust, it is a technical paper and it should be published in a more relevant journal (e.g. AMT). But this is up to the editor to decide.

The limitations are due to the aspect ratio range of the spheroids (0.7-1.4), in combination to the limited size range used in SphInX. As shown in Bi et al. (2018), the spheroids with aspect ratios of 0.7-1.4 present quite large particle depolarization values, larger than what is measured for dust particles (<0.3) (Fig. 1). These much larger values are decreased when considering smaller sizes for dust (as the ones considered in SphInX, with radius up to 2.2 $\mu$m), together with higher imaginary part for the refractive index, as the one retrieved with SphInX, with a value of 0.05. Figure 1 helps to explain the above, using the plots shown in Bi et al. (2018) for the particle depolarization ratio for two refractive indices with real part of 1.5 and imaginary parts of 0.01 and 0.1 (unfortunately, there is no plot in Bi et al. (2018) for the retrieved imaginary part of 0.05, but the values should be in between of what is shown in the plots in Fig. 1). As shown in the left plot in Fig. 1, the spheroids that are used in SphInX (red rectangular) will have particle depolarization ratio up to >0.8. The plot at the right shows that these values are greatly decreased for higher absorption (refractive index of 1.5+i0.1). In other words, forcing the retrieval to fit the depolarization measurements at 355, 532 and 1064nm at the certain aspect ratio range (which presents quite characteristic features for the particle depolarization ratio), in limited size range for dust (up to 2.2 $\mu$m), forces the imaginary part to values that do not indicate dust particles. The explanation provided in the manuscript of possible mixtures with smoke cannot stand under this light, without supporting measurements.

To conclude, the retrieved size distribution and refractive index of dust for the cases of SALTRACE campaign presented in the manuscript are most probably an artifact of the limited aspect ratio and size range used in SphInX retrieval algorithm. For this reason, there is no point in discussing these findings on a physical base, going further into characterizing the dust events during the SALTRACE campaign, but the paper should

go under major revisions, shifting its focus on the limitations of the current version of the SphInX retrieval algorithm for dust characterization.

Note 1: It is not surprising that the specific range of aspect ratios is excluded in the AERONET retrieval for dust (brown rectangulars in Fig. 1 and Fig. 11 in Dubovik et al. (2006)). See more details on this in my initial manuscript evaluation provided below.

Note 2: The authors did not include the fitting of the backscatter and extinction coefficients and the depolarization ratios at all wavelengths for both case studies, as requested in my initial manuscript evaluation. They are strongly advised to do so in the revised manuscript.

References:

Bi, L., Lin, W., Liu, D., and Zhang, K.: Assessing the depolarization capabilities of nonspherical particles in a super-ellipsoidal shape space, Opt. Express, 26, 1726–1742. https://doi.org/10.1364/OE.26.001726, 2018. Dubovik, O., Sinyuk, A., Lapyonok, T., Holben, B. N., Mishchenko, M., Yang, P., Eck, T. F., Voltne, H., Munoz, O., Veihelmann, B., Van der Zande, W. J., Leon, J.-F., Sorokin, M., and Slutsker, I.: Application of spheroid models to account for aerosol particle nonsphericity in remote sensing of desert dust, J. Geophys. Res., 111, D11208, doi:10.1029/2005JD006619, 2006.

THE INITIAL MANUSCRIPT EVALUATION

The retrieved aspect ratio distribution of the spheroidal particles presented in the manuscript is very different than the one presented in Dubovik et al. (2006) for dust particles (see Fig. 11 in Dubovik et al. (2006)). The backscattering is not included in the analysis of Dubovik et al. (2006), but this aspect ratio distribution has been shown to reproduce the backscatter measurements in other studies (see e.g. Lopatin et al. (2013)). The aspect ratio distribution shown in Dubovik et al. (2006) excludes the aspect ratios of 0.7-1.4. It is surprising that these are the aspect ratios that are shown to reproduce the dust measurements in your work. This result requires a more thorough

investigation. An explanation could be found in the work of Bi et al. (2018). There you can see that the particle depolarization for spheroids with aspect ratios of 0.7-1.4 presents much larger values than what is measured for dust particles (<0.3) (see Fig. 5a in Bi et al. (2018)). These much larger values are decreased when considering smaller sizes for dust (as the ones considered in your work, with radius up to 2.2 $\mu$m) and higher imaginary part for the refractive index, as the one you retrieve at $\sim$0.05 (see Fig. 5d in Bi et al. (2018)). Please go through the work of Bi et al. (2018) and provide a more thorough explanation for the retrieved aspect ratio distribution and imaginary part of the refractive index, considering the limited size range used in your work.

Moreover:

1. Use the same range for the x-axis in Fig. 5a and b and in Fig. 10a and b. 2. Provide plots with the fitting of the backscatter and extinction coefficients and the depolarization ratios at all wavelengths for both case studies.
* * *
[Figure]

**Figure 1**: Depolarization ratio (%) as a function of the aspect ratio and the size parameter of randomly oriented spheroids at two different refractive indices with the same real part: (c) 1.5 + i0.01 and (d) 1.5 + i0.1 (source: Fig. 5 in Bi et al. (2018)). The red rectangulars indicate the size and aspect ratio range used in SphInX retrieval algorithm, whereas the brown rectangulars indicate the size and aspect ratio range used in AERONET retrieval algorithm (Dubovik et al., 2006).

**Fig. 1.**

---

## Referee Comment (RC2) · Anonymous Referee #2 · 17 Jul 2020

The manuscript presents microphysical inversions of lidar data for two dust measurement cases during SALTRACE, using a non-spherical inversion scheme. Some qualitative comparisons are made with other retrieval methods.

I have three major concerns about the manuscript. First, it is rather difficult to understand its purpose. It is apparently more about the retrieval than about the characterization of the dust (in which case maybe better in AMT?); however, there is insufficient methodology to understand the retrieval and the comparisons are rather poor. A critical problem with the submission is that the retrieval seems to have serious flaws including at least two that the authors already believe contribute to the poor results, related to

the limitations in the LUT. I wonder if the severe underdeterminedness of the inversion is an even more fundamental problem with the retrieval, but I was unable to find a clear explanation of how the retrieval constrains the solution so I can't be sure. An exploration of this topic should also be undertaken. Finally, I am disappointed that the discussion of the results is not more candid. Quite significant differences in the comparisons are repeatedly dismissed as unimportant and described in vague but positive terms as "reasonable" or "qualitatively good", or more than once even "remarkable". Such lack of forthrightness does not serve science well. I encourage the authors to follow up on their findings about the shortcomings of the retrieval by trying to address them and improve the results, or, if that proves impossible, then learn why this retrieval approach is not working and write a thoughtful and candid report characterizing the issues and the lessons learned by the attempt.

Since I believe both the retrieval and manuscript need a significant overhaul before being resubmitted, I will pass over some details and focus mainly on more general points that might be helpful for a followup manuscript.

In the abstract (line 7) and elsewhere, since spheroidal particle databases have been used for dust microphysical retrievals for some time now, and since they do not use Mie theory, it would be better not to call this a new extension of Mie, but rather a new variation of a spheroidal particle retrieval with a different set of assumptions.

Repeatedly you refer to "the optical properties" (e.g. page 3, lines 6 and 9; page 10, line 24, and many other places). Please be more specific. What optical properties or measurements are you talking about?

In the abstract (line 16) and elsewhere, "qualitatively good agreement" is very vague and it's hard to actually agree that it is good, based on looking at your figures (e.g. Figs 7,11,13). Better to replace these vague statements with more precise description of the comparisons, whether good or poor, and then try to learn something from them. Even poor comparisons can be instructive and informative. Indeed you say on line 17

that this is a "fruitful" exercise. Be specific: what are the fruits of the exercise?

In the introduction (line 2-3, page 4) you discuss that the motivation for the new inversion is to eliminate the assumption of a fixed relationship between the aspect ratio of spheroids and their size that is employed by Dubovik et al (2006). The usual reason for making simplifying assumptions for an underconstrained retrieval is to add information to the system to enable making a retrieval. Doing this explicitly allows for assessing and making a judgement about the appropriateness of the assumptions. Without any assumed relationship between the shape and size, your retrieval must now retrieve an enormous amount of additional information. Since you have actually produced answers, you must have some other constraints or assumptions, either explicit or hidden. I could not determine from your methodology section what those additional constraints are, which leaves me unable to make an opinion about whether they are justified or not, but feeling somewhat pessimistic.

In section 2, the methodology, you describe the LUT with it's limited resolution and range, whose limitations you later hypothesize contribute to the inability of the retrieval to reproduce the results of other methods. Can extensions of the LUT be calculated? If that is impractical for some reason, are there other tests you can do to characterize the sensitivities to these limitations? For instance, forward calculations to understand how much error in the calculated observables is generated by interpolation in the RI grid? Experimenting with leaving out 355 nm measurements to extend the range of the particle radius?

I find the explanation at lines 5-15 of page 9 to be not enough information about the methodology of the retrieval. You have described a severely underdetermined problem with many more unknowns than measurements, but have not explained what constraints are used to "counteract the ill-posedness" (quoting from line 6). The three jargon-rich phrases at lines 8-10 are not very informative and the statement that you chose the Pade method because of its "superior behavior" is too vague and uninformative.

[Figure]

This section also contains quite a long passage relating to the much more specific question of how to compare with AERONET, accompanied by a flowchart. The text description is somewhat hard to follow. The flowchart does not help me at all, partly because there is no clear indication how the different boxes are meant to relate to each other. Please give, in the text, the actual equations that explicitly describe the relationships between the quantities of interest and which you used in the comparison. This would be more direct and universally understandable. (Furthermore, the flow would be improved if the retrieval methodology came before the methodology relating to the much more specific question of how to compare with AERONET.)

In section 3, the microphysics results, P12 line 6 states that the variability in the results between different parts of the retrievals is low, followed by a pair of sentences giving a variety of different possible conclusions we could draw from this (1) there actually is a small amount of variability in the aerosol (2) the data are of high quality (3) the smoothing on the input data reduced the apparent variability compared to the true variability of the aerosol (4) the greater information content in the lidar measurements (8 lidar measurements at each level) somehow reduces the variability in the retrieved output. These strike me as all very different explanations. While they all may in fact be true simultaneously, there is very little support for any of them and no further attempt to distinguish between the hypothesis with other experiments (such as reducing the number of input channels, reducing the smoothing, relaxing the assumptions in the retrieval), etc., so I don't know what's being learned here. In general, hypothesis need some testing and support, whether from actual experiments or "thought experiments". Also, given that I could not find answers to my questions about the assumptions or constraints in the retrieval methodology (and given the poor agreement with other methods) I worry that another possible explanation for low variability in the output might be that the retrieval is over-constrained. Can you demonstrate that this is not a problem?

P13, line 8-11 calls out what is described as "a shortcoming of our approach". This should be addressed further by attempting to correct the approach, or if that is not

possible, then design an experiment to quantify the sensitivity of the results to the resolution of the refractive index grid.

Similarly P13, line 30 indicates that the inversion has a size cutoff that appears to be significantly impacting the retrieval results such that it is unable to produce a good comparison with the other methods. This seems likely to me, but critical followup is missing. What kind of sensitivity study or other data is there to support your hypothesis that this is the primary reason for the differences? What assurance do you have that there are not other problems? And most importantly, if you already know what's wrong with the retrieval, probably it should be fixed before publication.

In section 3.1.3 discussing the comparisons, these comparisons are really rather poor, but they are described as "reasonable agreement" "remarkable" and "same qualitative behavior". These are all rather vague and even a bit misleading. What does "same qualitative behavior" mean? Do you mean to say the vertical profile shapes are correlated? If so, go ahead and check it out using a regression. Perhaps you will indeed find that even though the slope is far off the correlation is high. If so, you can make that conclusion and then try to figure out what it tells you about your retrieval (i.e. why would it produce the right shape but be biased so badly?) However, looking at it purely visually, I don't see the good correlation; except for the transition near 3 km, none of the peaks in the two profiles appear to be at the same altitudes, for instance. Next, you describe the fine mode difference as "only" 5.5 cubic microns per cubic cm. But this is not a small difference; as a fraction of the fine mode volume concentration it's enormous. Also, it's not a random difference averaging to to that value, it's a systematic difference across the whole profile, and with a distinct altitude dependence. In other words, there's really nothing to convince me that these two retrievals are describing the same thing. You discuss a possible explanation for differences in the coarse mode, but what could be the problem with the fine mode? If you have a hypothesis, state it and test it (or even better, use it to fix the problem). I have similar concerns about the comparisons in section 3.2.2.

In section 3.2.3 (comparisons with FALCON), there is again a statement (or perhaps implication) of an unsupported hypothesis that the differences after adjusting for the cutoff are primarily due to space and time differences between the measurements. But if that were so, wouldn't the POLIPHON results be equally impacted and therefore show poor agreement with the insitu measurements?

Figures 5 and 11 for the comparison with AERONET make it quite clear that the 2.2 micron cutoff is a severe handicap of the retrieval, a significant constraint which is not in line with reality. It should be straightforward to do a forward calculation to determine by what percentage of lidar backscatter and extinction are missing if the AERONET distribution were cut off at 2.2 microns; it seems likely that this will be far more than the discrepancy you have chosen to allow in the inversion.

In the text of section 3.3 the statement that the volume concentrations are "within the same order of magnitude" is mystifying to me. I agree that the calculations at lines 30-31 (pg 15) as a way of estimating the column values from the profile are fine, but the statement refers to the figures, where two different quantities with different dimensions and units are shown. How can quantities with different dimensions be described as "within the same order of magnitude"? Better to show the converted quantities on the figures, with the AERONET dimensions of volume per volume times height, and in the same units, so they can be directly compared.

The final statement in the section about good agreement between the sun-photometer and POLIPHON is really very odd, since it completely leaves out the Sphinx retrieval that this manuscript is about. It appears again that the authors are very shy about describing less-than-desirable results of the comparisons.

One additional more specific point: On page 2, lines 3-14, even though it is properly put into quotation marks and attributed, I'm not sure it's appropriate to have a paragraph-long word-for-word quote from another source. Of course, that's an editorial decision, but in any case I think it would be better to extract from this quote your own view

on what are the important points and paraphrase them in your own words (while still referencing the original source, of course).

---

## Referee Comment (RC3) · Anonymous Referee #1 · 20 Aug 2020

The authors present a new method to simultaneously infer the size distribution and the aspect ratio distribution of non-spherical mineral dust particles from advanced li-dar measurements. The scope of this approach is good as other retrievals are often limited from the outset in their general applicability by assuming a fixed aspect ratio distribution. However, the authors clearly show that the current version of the retrieval with a very limited parameter space considered in its look-up table is not yet up to the task of dealing with real-life data. I recommend to reject this contribution for publication in ACP as it is too technical to be within the scope of the journal and shows strong deficiencies in scientific and presentation quality. The authors should thoroughly revise their work into a technical note or paper, e.g. for AMT, in which they can clearly lay out

the novelties of their method as well as the limitations that still need to be overcome.

The other Referees have already provided a wealth of detailed comments that cover most of my concerns with this contribution. In addition, the authors need to improve the structure of their presentation to allow the reader to follow their work. For instance, it would be good to first introduce the available measurements and later explain how they will be used in the description of the retrieval. Several figures are presented and never discussed (Figs. 3 and 9) or discussed long after referencing to them first (Figs. 5, 7, 11, 13). Some parameters get acronyms that are used only once while others get two variables. Comparisons are entirely qualitative, much to optimistic and mostly mixing up different things or incomparable studies (apples and oranges comparisons).